# GraphQA: Protein Model Quality Assessment using Graph Convolutional Network

**Federico Baldassarre[1], Hossein Azizpour[1], David Menéndez Hurtado[2], Arne Elofsson[2]**
[1] KTH - Royal Insitute of Technology
[2] Stockholm University, Science for Life Laboratory and Dep. of Biochemistry and Biophysics

## Abstract

Proteins are ubiquitous molecules whose function in biological processes is determined by their 3D structure. Experimental identification of a protein's structure can be time-consuming, prohibitively expensive, and not always possible. Alternatively, protein folding can be modeled using computational methods, which however are not guaranteed to always produce optimal results.

GraphQA is a graph-based method to estimate the quality of protein models, that possesses favorable properties such as representation learning, explicit modeling of both sequential and 3D structure, geometric invariance and computational efficiency. In this work, we demonstrate significant improvements over the state-of-the-art for both hand-engineered and representation-learning approaches, as well as carefully evaluating the individual contributions of GraphQA components.

## 1 Introduction

Protein molecules are predominantly present in biological forms, responsible for their cellular functions. Therefore, understanding, predicting and modifying proteins in biological processes are essential for medical, pharmaceutical and genetic research. Such studies strongly depend on discovering mechanical and chemical properties of proteins through the determination of their structure.

At the high level, a protein molecule is a chain of hundreds of smaller molecules called amino acids. Identifying a protein's amino-acid sequence is nowadays straightforward. However, the function of a protein is primarily determined by its 3D structure. Spatial folding can be determined experimentally, but the existing procedures are time-consuming, prohibitively expensive and not always possible. Thus, several computational techniques were developed for protein structure prediction (Arnold et al., 2006; Wang et al., 2017; Xu, 2019). So far, no single method is always best, e.g. some protein families are best modeled by certain methods, also, computational methods often produce multiple outputs. Therefore, candidate generation is generally followed by an evaluation step. This work focuses on Quality Assessment (QA) of computationally-derived models of a protein (Lundstrom et al., 2001; Won et al., 2019).

QA, also referred to as model accuracy estimation (MAE), estimates the quality of computational protein models in terms of divergence from their native structure. The downstream goal of QA is two-fold: to find the best model in a pool of models and to refine a model based on its local quality.

Computational protein folding and design have recently received attention from the machine learning community (Wang et al., 2017; Xu, 2019; Jones & Kandathil, 2018; Ingraham et al., 2019b; Anand & Huang, 2018; Evans et al., 2018; AlQuraishi, 2019), while QA has yet to follow. This is despite the importance of QA for structural biology and the availability of standard datasets to benchmark machine learning techniques, such as the biannual CASP event (Moult et al., 1999). The field of bioinformatics, on the other hand, has witnessed noticeable progress in QA for more than a decade: from earlier works using artificial neural networks (Wallner & Elofsson, 2006) or support vector machines (Ray et al., 2012; Uziela et al., 2016) to more recent deep learning methods based on 1D-CNNs, 3D-CNNs and LSTMs (Hurtado et al., 2018; Derevyanko et al., 2018; Pagès et al., 2018; Conover et al., 2019).

In this work, we tackle Quality Assessment with Graph Convolutional Networks, which offer several desirable properties over previous methods. Through extensive experiments, we show significant improvements over the state-of-the-art, and offer informative qualitative and quantitative analyses.

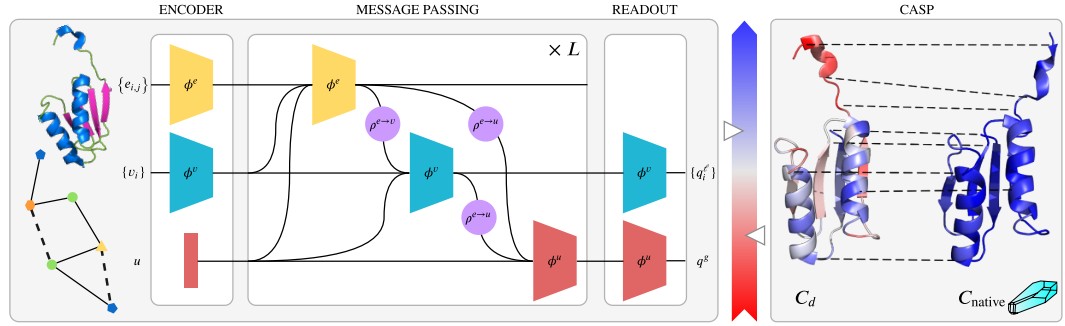

Figure 1: **Protein Quality Assessment.** GRAPHQA predicts local and global scores from a protein's graph using message passing among residues with chemical bond or spatial proximity. CASP QA algorithms score protein models by comparison with experimentally-determined conformations.

## 1.1 RELATED WORKS

**Protein Quality Assessment (QA)** methods are evaluated in CASP (Moult et al., 1995) since CASP7 (Cozzetto et al., 2007). Current techniques can be divided into two categories: single-model methods which operate on a single protein model to estimate its quality (Wallner & Elofsson, 2003), and consensus methods that use consistency between several candidates to estimate their quality (Lundstrom et al., 2001). Single-model methods are applicable to a single protein in isolation and in the recent CASP13 performed comparable to or better than consensus methods for the first time (Cheng et al., 2019). Recent single-model QA works are based on deep learning, except VoroMQA that takes a statistical approach on atom-level contact area (Olechnovic & Venclovas, 2017). 3DCNN adopts a volumetric representation of proteins (Derevyanko et al., 2018). Ornate improves 3DCNN by defining a canonical orientation (Pagès et al., 2018). ProQ3D uses a multi-layer perceptron on fixed-length protein descriptors (Uziela et al., 2017). ProQ4 adopts a pre-trained 1D-CNN that is fine-tuned in a siamese configuration with a rank loss (Hurtado et al., 2018). VoroMQA and ProQ3D are among the top performers of CASP13 (Won et al., 2019).

**Graph Convolutional Networks (GCNs)** bring the representation learning power of CNNs to graph data, and have been recently applied with success to multiple domains, e.g. physics (Gonzalez et al., 2018), visual scene understanding (Narasimhan et al., 2018) and natural language understanding (Kipf & Welling, 2017). Molecules can be naturally represented as graphs and GCNs have been proven effective in several related tasks, including molecular representation learning (Duvenaud et al., 2015), protein interface prediction (Fout et al., 2017), chemical property prediction (Niepert et al., 2016; Gilmer et al., 2017; Li et al., 2018a), drug-drug interaction (Zitnik et al., 2018), drug-target interaction (Gao et al., 2018) molecular optimization (Jin et al., 2019), and generation of proteins, molecules and drugs (Ingraham et al., 2019a; You et al., 2018; Liu et al., 2018; Li et al., 2018b; Simonovsky & Komodakis, 2018). However, to the best of our knowledge, GCNs have never been applied to the problem of protein quality assessment.

## 1.2 CONTRIBUTIONS

- This work is the first to tackle QA with GCN which bring several desirable properties over previous methods, including representation learning (3DCNN, Ornate), geometric invariance (VoroMQA, Ornate), sequence learning (ProQ4, AngularQA), explicit modeling of 3D structure (3DCNN, Ornate, VoroMQA) and computational efficiency.
- Thanks to these desirable properties, a simple GCN setup achieves improved results compared to the more sophisticated state-of-the-art methods such as ProQ4. This is demonstrated through extensive experiments on multiple datasets and scoring regimes.
- Novel representation techniques are employed to explicitly reflect the sequential (residue separation) and 3D structure (angles, spatial distance and secondary structure) of proteins.
- Enabled by the use of GCN, we combine the optimization of local and global score for QA, improving over the performance of a global-only scoring method.
- Through an extensive set of ablation studies, the significance of different components of the method, including architecture, loss, and features, are carefully analyzed.

PyTorch implementation, datasets and experiments: github.com/baldassarreFe/protein-quality-gn.

## 2 METHOD

We start describing our method by arguing for representation of protein molecules as graphs in learning tasks, then we define the problem of protein quality assessment (QA), and finally we present the proposed GRAPHQA architecture.

### 2.1 PROTEIN REPRESENTATION AS GRAPHS

Proteins are large molecular structures that perform vital functions in all living organisms. At the chemical level, a protein consists of one or more chains of smaller molecules, which we interchangeably refer to as **residues** for their role in the chain, or as **amino acids** for their chemical composition. The sequence of residues $S = \{ a_i \}$ that composes a protein represents its *primary structure*, where $a_i$ is one of the 22 amino acid types. The interactions between neighboring residues and with the environment dictate how the chain will fold into complex spatial structures that represent the protein's *secondary structure* and *tertiary structure*.

Therefore, for learning tasks involving proteins, a suitable representation should reflect both the identity and sequence of the residues, i.e. the primary structure, and geometric information about the protein's arrangement in space, i.e. its tertiary structure. Some works use RNN or 1D-CNN to model proteins as sequence with the spatial structure potentially embedded in the handcrafted residue features (Hurtado et al., 2018; Conover et al., 2019). Other recent works explicitly model proteins' spatial structure using 3D-CNN but ignore its sequential nature (Derevyanko et al., 2018; Pagès et al., 2018). We argue that a graph-based learning can explicitly model both the sequential and geometric structures of proteins. Moreover, it accommodates proteins of different lengths and spatial extent, and is invariant to rotations and translations.

In the simplest form, a protein can be represented as a linear graph, where nodes represent amino acids and edges connect consecutive residues according to the primary structure. This set of edges, which represent the covalent **bonds** that form the protein backbone, can be extended to include the interactions between non-consecutive residues, e.g. through Van der Waals forces or hydrogen bonds, commonly denoted as **contacts**. By forming an edge between all pairs of residues that are within a chemically reasonable distance of each other, the graph becomes a rich representation of both the sequential and geometric structure of the protein (figure 2). We refer to this representation, composed of residues, bonds and contacts, as the **protein graph**:

$$\mathcal{P} = \Big( \{ \, \boldsymbol{v}_i \, \}, \quad \{ \, \boldsymbol{e}_{i,j}^{\text{bond}} \mid |i - j| = 1 \, \} \cup \{ \, \boldsymbol{e}_{i,j}^{\text{contact}} \mid |i - j| > 1, \|C_i - C_j\| \leq d_{\max} \, \} \Big), \quad (1)$$

where $i, j = 1, \ldots, |S|$ are residue indices, $C = \{ \, (x, y, z)_i \, \}$ represents the spatial arrangement of the residues, i.e. the protein's **conformation**, and $d_{\max}$ is a cutoff distance for contacts.

With the protein's structure encoded in the graph, additional residue and relationship features can be encoded as nodes and edges attributes, $\boldsymbol{v}_i$ and $\boldsymbol{e}_{i,j}$ respectively. Section 3.2 describes, in detail, an attribution that preserves the sequence information and 3D geometry while remaining invariant to rotation.

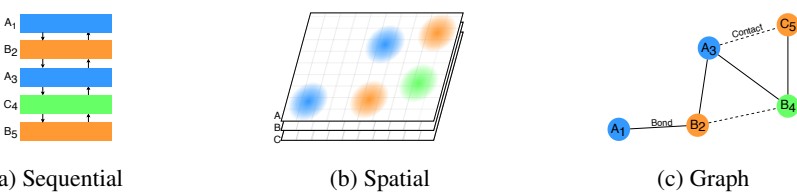

|  (a) Sequential  |  (b) Spatial  |  (c) Graph  |

Figure 2: **Protein representations for learning.** Sequential representations for LSTM or 1D-CNN fail to represent spatial proximity of non-consecutive residues. Volumetric representations for 3D-CNN fail instead to capture sequence information and is not rotation invariant. Protein graphs explicitly represent both sequential and spatial structure, and are geometrically invariant by design.

## 2.2 PROTEIN QUALITY ASSESSMENT

Experimental identification of a protein's **native structure** can be time consuming and prohibitively expensive. Alternatively, computational folding methods are used to generate **decoy** conformations for a specific **target** protein. Since no single method is consistently best, a Quality Assessment (QA) step is used to identify the decoys that most correctly represent the native structure.

If the native structure $C_{\text{native}}$ is experimentally determined, the quality of a decoy can be measured by comparison, e.g., in the CASP challenge, decoys submitted for a target are scored against the unreleased native structure. Some comparative algorithms compute global (per decoy) scores, which can be used for ranking and represent the principal factor for CASP, while others produce local (per residue) scores which help identify incorrect parts of a decoy (Uziela et al., 2018).

In most scenarios, however, the native structure is not available and quality must be estimated based on physical and chemical properties of the decoy, e.g. in drug development, it would be unpractical to synthesize samples of novel proteins and researchers rely on computational folding and quality assessment instead.

Here we introduce GRAPHQA, a graph-based QA neural network that learns to predict local and global scores, with minimal feature and model engineering, using existing datasets of scored proteins. In this paper, we train GRAPHQA on two widely-used scoring algorithms: the Global Distance Test Total Score (Zemla, 2003), which is the official CASP score for decoy-level quality assessment, and the Local Distance Difference Test (Mariani et al., 2013), a residue-level score. We denote them as $q^g := \text{GDT\_TS}(C^d, C^{\text{native}})$ and $\{ q_i^\ell \} := \text{LDDT}(C^d, C^{\text{native}})$.

With $\text{GRAPHQA}_i^\ell(\mathcal{P})$ and $\text{GRAPHQA}^g(\mathcal{P})$ denoting the network's local and global predictions for an input $\mathcal{P}$, the learning objective is to minimize the following Mean Squared Error (MSE) losses:

$$\mathcal{L}_\ell = \sum_i^{|S|} \left[ \text{GRAPHQA}_i^\ell(\mathcal{P}) - q_i^\ell \right]^2 \qquad \mathcal{L}_g = \left[ \text{GRAPHQA}^g(\mathcal{P}) - q^g \right]^2 . \qquad (2)$$

Note that, for the sole purpose of sorting decoy according to ground-truth quality, training with a ranking loss would be sufficient (Derevyanko et al., 2018). Instead, MSE forces the output to match the quality score, which is a harder objective, but results in a network can be more easily inspected and possibly used to improve existing folding methods in an end-to-end fashion (section 4.2).

## 2.3 GRAPHQA ARCHITECTURE

GRAPHQA is a graph convolutional network that operates on protein graphs using the message-passing algorithm described in Battaglia et al. (2018). The building block of GRAPHQA, a graph layer, takes a protein graph as input (with an additional global feature $\boldsymbol{u}$), and performs the following propagation steps to output a graph with updated node/edge/global features and unchanged structure:

$$\begin{aligned}
\boldsymbol{e}_{i,j}' &= \phi^e\left(\boldsymbol{e}_{i,j}, \boldsymbol{v}_i, \boldsymbol{v}_j, \boldsymbol{u}\right) & \text{Update edges} \\
\bar{\boldsymbol{e}}_i' &= \rho^{e\rightarrow v}\left(\{\boldsymbol{e}_{j,i}'\}\right) & \text{Aggregate edges} \\
\boldsymbol{v}_i' &= \phi^v\left(\bar{\boldsymbol{e}}_i', \boldsymbol{v}_i, \boldsymbol{u}\right) & \text{Update nodes}
\end{aligned} \quad \Bigg| \quad \begin{aligned}
\bar{\boldsymbol{e}}' &= \rho^{e\rightarrow u}\left(\{\boldsymbol{e}_{i,j}'\}\right) & \text{Aggregate all edges} \\
\bar{\boldsymbol{v}}' &= \rho^{v\rightarrow u}\left(\{\boldsymbol{v}_i'\}\right) & \text{Aggregate all nodes} \\
\boldsymbol{u}' &= \phi^u\left(\bar{\boldsymbol{e}}', \bar{\boldsymbol{v}}', \boldsymbol{u}\right) & \text{Update global features}
\end{aligned}$$

where $\phi$ represent three update functions that transform nodes/edges/global features (e.g. a MLP), and $\rho$ represent three pooling functions that aggregate features at various levels (e.g. sum or mean).

Similarly to CNNs, multiple graph layers are stacked together to allow local information to propagate to increasingly larger neighborhoods (i.e. receptive field). This enables the network to learn quality-related features at multiple scales: secondary structures in the first layers, e.g. $\alpha$-helices and $\beta$-sheets, and larger structures in deeper layers e.g. domain structures and arrangements.

The GRAPHQA architecture is conceptually divided in three stages (figure 1). At the input, the **encoder** increases the node and edge features' dimensions through $2\times$(Linear-Dropout-ReLU) transformation and adds a global bias. Then, at its the core, $L$ **message-passing** layers operate on the encoded graph, leveraging its structure to propagate and aggregate information. The update functions $\phi$ consist of Linear-Dropout-ReLU transformations, with the size of the linear layers progressively decreasing. We use average pooling for the aggregation functions $\rho$, since preliminary experiments with max/sum pooling performed poorly. Finally, the **readout** layer outputs local and global quality scores by applying a Linear-Sigmoid operation to the latest node and global features, respectively.

## 3 EXPERIMENTS

### 3.1 EXPERIMENTAL SETUP

Following the common practice in Quality Assessment, we use the data from past years' editions of CASP, encompassing several targets with multiple scored decoys each. Removing all sequences with $|S| < 50$ from CASP 7-10 results in a dataset of $\sim$100k scored decoys $(\mathcal{P}, \{ q_i^\ell \}, q^g)^{t,d}$, which we randomly split into a training set (402 targets) and a validation set for hyper-parameter optimization (35 targets). CASP 11 and 12 are set aside for testing against top-scoring methods (table 3).

We evaluate the performances of GRAPHQA on the following standard metrics. At the global level, we compare the predicted and ground-truth GDT_TS scores and compute: Root Mean Squared Error (RMSE), Pearson correlation coefficient computed across all decoys of all targets ($R$), and Pearson correlation coefficient computed on a per-target basis and then averaged over all targets ($R_{target}$). At the local level, we compare the predicted and ground-truth LDDT scores and compute: RMSE, Pearson correlation coefficient computed across all residues of all decoys of all targets ($R$), and Pearson correlation coefficient computed on a per-decoy basis and then averaged over all decoys of all targets ($R_{model}$). Of these, we focus on $R_{target}$ and $R_{model}$, which respectively measure the ability to rank decoys by quality and to distinguish the correctly-predicted parts of a model from those that need improvement. A detailed description of these and other metrics can be found in appendix E.

### 3.2 FEATURES

**Node features** The node attributes $\boldsymbol{v}_i$ of a protein graph $\mathcal{P}$ represent the identity, statistical, and structural features of the $i$-th residue. We encode the residue identity using a one-of-22 encoding of the corresponding amino acid. Following Hurtado et al. (2018), we also add two residue-level statistics computed using Multiple Sequence Alignment (MSA) (Rost et al., 1994), namely *self-information* and *partial entropy*, each described by a 23-dimensional vector. Finally, we add a 14-dimensional vector of 3D spatial information including the dihedral angles, surface accessibility and secondary structure type as determined by DSSP (Kabsch & Sander, 1983).

**Edge features** An edge represents either a contact or a bond between two residues $i$ and $j$ w.r.t. to the conformation $C = \{ (x, y, z)_i \}$. An edge always exists between two consecutive residues, while non-consecutive residues are only connected if $||C_i - C_j|| < d_{max}$, with $d_{max}$ optimized on the validation set. We further enrich this connectivity structure by encoding spatial and sequential distances as an 8D feature vector $\boldsymbol{e}_{i,j}$. Spatial distance is encoded using a radial basis function $\exp(-d_{i,j}^2/\sigma)$, with $\sigma$ determined on the validation set. Sequential distance is defined as the number of amino acids between the two residues in the sequence and expressed using a **separation encoding**, i.e. a one-hot encoding of the separation $|i - j|$ according to the classes $\{ 0, 1, 2, 3, 4, 5 : 10, > 10 \}$.

### 3.3 OPTIMIZATION AND HYPERPARAMETER SEARCH

The MSE losses in equation 2 are weighted as $\mathcal{L}_{tot} = \lambda_\ell \mathcal{L}_\ell + \lambda_g \mathcal{L}_g$ and minimized using Adam Optimizer (Kingma & Ba, 2014) with $L_2$ regularization. GRAPHQA is significantly faster to train than LSTM or 3D-CNN methods, e.g. 35 epochs takes $\sim$2 hours on one NVIDIA 2080Ti GPU with batches of 200 graphs. This allows us to perform extensive hyper-parameter search. Table 4 reports the search space, as well as the parameters of the model with highest $R_{target}$ on the validation set.

## 4 EVALUATION

We compare GRAPHQA with the following methods, chosen either for their state-of-the-art performances or because they represent a class of approaches for Quality Assessment. ProQ3D (Uziela et al., 2017) computes fixed-size statistical descriptions of the decoys in CASP 9-10, including Rosetta energy terms, which are then used to train a Multi Layer Perceptron on quality scores. In ProQ4 (Hurtado et al., 2018), a 1D-CNN is trained to predict LDDT scores from a vectorized representation of protein sequences, a global score is then obtained by averaging over all residues. ProQ4 is pretrained on a large dataset of protein secondary structures and then fine tuned on CASP 9-10 using a siamese configuration to improve ranking performances. Their results are reported on both CASP 11, which is used as a validation set, and CASP 12. 3DCNN (Derevyanko et al., 2018) trains

Table 1: **Comparison of state-of-the-art QA methods.** At the residue level we compare LDDT scores and report Pearson correlation and Pearson correlation per model. At the global level we compare GDT_TS scores and report Pearson correlation and Pearson correlation per target.

| | CASP 11 | | | | CASP 12 | | | |
| | GDT_TS | | LDDT | | GDT_TS | | LDDT | |
| | $R$ | $R_{\text{target}}$ | $R$ | $R_{\text{model}}$ | $R$ | $R_{\text{target}}$ | $R$ | $R_{\text{model}}$ |
|---|---|---|---|---|---|---|---|---|
| ProQ3D | .772 | .452 | .84 | **.61** | .806 | .609 | | |
| ProQ4 | | | .77 | .56 | | | .772 | .516 |
| VoroMQA | .651 | .457 | | | .605 | .559 | | |
| Rwplus | | .206 | | | -.096 | .417 | | |
| AngularQA | .651 | .439 | | | | | | |
| 3D CNN | .629 | .421 | | | | .607 | | |
| Ornate | .637 | .386 | | | .670 | .491 | | |
| GRAPHQA | **.910** | **.740** | **.855** | **.610** | **.843** | **.745** | **.843** | **.573** |
| GRAPHQA$_{\text{RAW}}$ | .836 | .609 | .799 | .529 | .816 | .673 | .796 | .507 |

a CNN on a three-dimensional representation of atomic densities to rank the decoys in CASP 7-10 according to their GDT_TS scores. Notably, no additional feature is used other than atomic structure and type, however, the fixed-size volumetric representation of this method is sensitive to rotations and does not scale well with protein size. Ornate (Pagès et al., 2018) applies a similar 3D approach to predict local CAD-scores (Olechnovic & Venclovas, 2017) and achieves rotation invariance by specifying a canonical residue-centered orientation. Although optimized for local scoring, the average of the predicted scores is shown to correlate well with GDT_TS. AngularQA (Conover et al., 2019) feeds a sequence-like representation of the protein structure to an LSTM to predict GDT_TS scores. The LSTM network is trained on decoys from 3DRobot and CASP 9-11, while CASP 12 is used for model selection and testing. VoroMQA and RWplus (Olechnovič & Venclovas, 2017; Zhang & Zhang, 2010) are two statistical potential methods that represent an alternative to the other machine-learning based methods.

Table 1 compares the performances of GRAPHQA and other state-of-the-art methods on CASP 11 and 12, while figure 3 contains a graphical representation of true vs. predicted scores for all target in CASP 12, and an example funnel plot for the decoys of a single target. A more in-depth evaluation on the stage 1 and stage 2 splits of CASP 11, 12, 13 and CAMEO can be found in appendix F.

Of all methods, only GRAPHQA and ProQ4 co-optimize for local and global predictions, the former thanks to the graph-based architecture, the latter thanks to its siamese training configuration (the results reported for ProQ3D refer to two separate models trained for either local or global scores). At the local level, our method proves to be on par or better than ProQ3D and ProQ4, demonstrating the ability to evaluate quality at the residue level and distinguishing correctly predicted parts of the protein chain. At the global level, significantly higher $R$ and $R_{\text{target}}$ metrics indicate than GRAPHQA is more capable than other state-of-the-art methods at ranking decoys based on their overall quality. As shown in our ablation studies (section 5), hand-engineered features like MSA and DSSP contribute to the performances of GRAPHQA, yet we wish to prove that our method can learn directly from raw data. GRAPHQA$_{\text{RAW}}$ is a variant that relies uniquely on the one-hot encoding of amino acid identity, similarly to how 3D-CNN and Ornate employ atomic features only. The results for GRAPHQA$_{\text{RAW}}$ show that, even without additional features, our method outperforms purely representation-learning methods.

## 4.1 ABLATION STUDIES

In this section we analyse how various components of our method contribute to the final performance, ranging from optimization and architectural choices to protein feature selection. Unless stated otherwise, all ablation studies follow the training procedure described in section 3.3 for a lower number of epochs. We report results on CASP 11 as mean and standard deviation of 10 runs.

**Local and global co-optimization** We investigate the interplay between local and global predictions, specifically whether co-optimizing for both is beneficial or detrimental. At the global level, models trained to predict only global scores achieve a global RMSE of $0.129\pm.007$, whereas models trained to predict both local and global scores obtain $0.117\pm.006$, suggesting that local scores can

provide additional information and help the assessment of global quality. At the local level instead, co-optimization does not seem to improve performances: models trained uniquely on local scores achieve a local RMSE of $0.121\pm.002$, while models trained to predict both obtain $0.123\pm.004$.

**Connectivity and Architecture** In this study, we test the combined effects of the depth of the network $L$ and the cutoff value $d_{\max}$. On the one hand, every additional message-passing layer allows to aggregate information from a neighborhood that is one hop larger than the previous, effectively extending the receptive field at the readout. On the other hand, the number of contacts included in the graph affects its connectivity and the propagation of messages, e.g. low $d_{\max}$ correspond to a low average degree and long shortest paths between any two residues, and vice versa (section B.2).

Thus, an architecture that operates on sparsely-connected graphs will require more message-passing layers to achieve the same holistic view of a shallow network operating on denser representations. However, this trade off is only properly exposed if $\boldsymbol{u}, \phi^u, \rho^u$ are removed from the architecture. In fact, this global pathway represents a propagation shortcut that connects all nodes in the graph and sidesteps the limitations of shallow networks. With the global pathway disabled, global predictions are computed in the readout layer by aggregating node features from the last MP layer.

Figure 4 reports the RMSE obtained by networks of different depth with no global path, operating on protein graphs constructed with different cutoff values. As expected, the shallow 3-layer architecture requires more densely-connected inputs to achieve the same performances of the 9-layer network. Surprisingly, local predictions seem to be more affected by these factors than global predictions, suggesting that a large receptive field is important even for local scores.

**Node and Edge Features** We evaluate the impact of node and edge features on the overall prediction performances (figure 5). For the nodes, we use the amino acid identity as a minimal representation and combine it with: a) DSSP features, b) partial entropy, c) self information, d) both DSSP and MSA features. All features improve both local and global scoring, with DSSP features being marginally more relevant for LDDT. For the edges, we evaluate the effect of having either: a) a binary indicator of bond/contact, b) geometric features, i.e. the euclidean distance between residues, c) sequential features, i.e. the categorical encoding of the separation between residues, d) both distance and separation encoding. Progressively richer edge features seem to be benefit LDDT predictions, while little improvement can be seen at the global level.

## 4.2 VISUALIZATION AND EXPLAINABILITY

The design of GRAPHQA makes it suitable not only for scoring, but also to identify refinement opportunities for computationally-created decoys. Figure 6 shows a decoy that correctly models the native structure of its target for most of the sequence, but one extremity to which both GRAPHQA and LDDT assign low local scores. Unlike LDDT, however, GRAPHQA is fully differentiable and the trained model can be used to explain the factors that influenced a low score and provide useful feedback for computational structure prediction.

A simple approach for explaining predictions of a differentiable function $f(\boldsymbol{x})$ is Sensitivity Analysis (Baehrens et al., 2010; Simonyan et al., 2014), which uses $\|\nabla_{\boldsymbol{x}} f\|$ to measure how variations in the input affect the output. In figure 6 we consider the scores predicted for two different residues and compute the magnitude of the gradients w.r.t. the edges of the graph. Interestingly, GRAPHQA

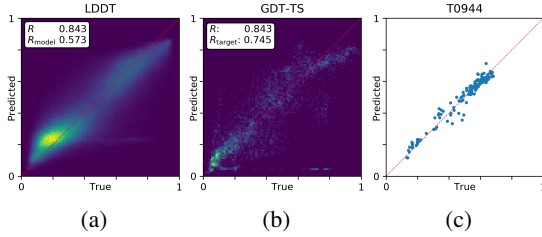

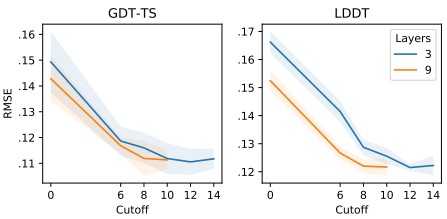

Figure 3: Histograms of true vs. predicted LDDT (a) and GDT_TS (b) scores on CASP 12, (c) funnel plot of the decoys of target T0944 (PDB 5ko9).

Figure 4: Trade-off between number of message-passing layers and connectivity of the protein graph.

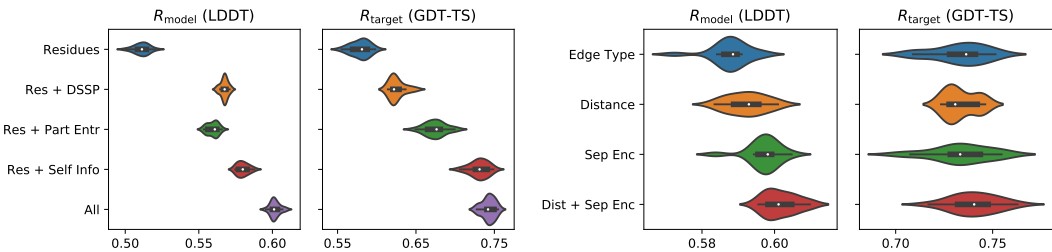

Figure 5: **Ablation study of node and edge features.** All node features improve both local and global scoring, with DSSP features being marginally more relevant for LDDT (left). Richer edge features benefit LDDT predictions, while little improvement can be seen at the global level (right).

is able to capture quality-related dependencies not only in the neighborhood of the selected residues, but also further apart in the sequence.

Finally, we measure whether the global predictions of GRAPHQA could be used to improve the contact maps used by computational methods to build protein models. If the network has learned a meaningful scoring function for a decoy, then the gradient of the score w.r.t. the contact distances should aim in the direction of the native structure. Considering all decoys of all targets in CASP 11, we obtain an average cosine similarity $\cos\left(\partial \mathrm{GRAPHQA}^g/\partial d, d_{\mathrm{decoy}} - d_{\mathrm{native}}\right)$ of $0.14\pm.08$, which suggests that gradients can be used as a coarse feedback for end-to-end protein model prediction.

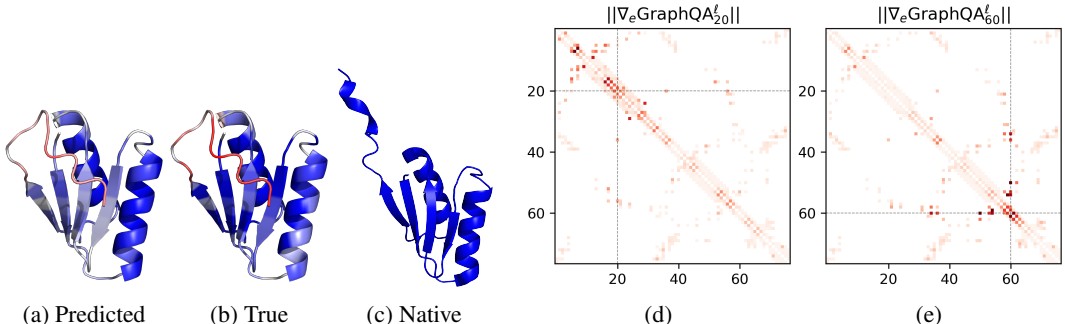

Figure 6: **One decoy of T0773 in CASP11.** Both GRAPHQA (a) andLDDT (b) assign low local scores to a segment of the decoy (in red), highlighting a discrepancy w.r.t. the native structure (c). The gradient magnitude w.r.t. the edges of the predicted LDDT score for residues 20 (d) and 60 (e) reveal long range dependencies inside the protein graph.

## 5    CONCLUSION

For the first time we applied graph convolutional networks to the important problem of protein quality assessment (QA). Since proteins are naturally represented as graphs, GCN allowed us to collect the individual benefits of the previous QA methods including representation learning, geometric invariance, explicit modeling of sequential and 3D structure, simultaneous local and global scoring, and computational efficiency. Thanks to these benefits, and through an extensive set of experiments, we demonstrated significant improvements upon the state-of-the-art results on various metrics and datasets and further analyzed the results via thorough ablation and qualitative studies.

Finally, we wish that Quality Assessment will gain popularity in the machine learning community, that could benefit from several curated datasets and ongoing regular challenges. We believe that richer geometric representations, e.g. including relative rotations, and raw atomic representations could represent an interesting future direction for learning-based Quality Assessment.

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

# A    PROTEIN QUALITY ASSESSMENT

For the interested reader, we describe here in more detail how the Global Distance Test Total Score (Zemla, 2003) and the Local Distance Difference Test (Mariani et al., 2013) are computed. Furthermore, we provide an intuition over what the benefits and downsides of each method are and motivate why a better quality assessment should consider both a global measure and a local measure.

**Global Distance Test Total Score (GDT_TS)** Global Distance Test Total Score (GDT_TS) is a global-level score obtained by first superimposing the structure of a decoy to the experimental structure using an alignment heuristic, and then computing the fraction of residues whose position is within a certain distance from the corresponding residue in the native structure (figure 7). This percentage is computed at different thresholds and then averaged to produce a score in the range $[0, 100]$, which we rescale between 0 and 1 (table 2).

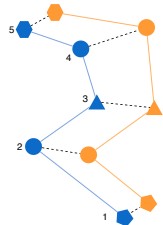

Figure 7: GDT_TS

Table 2

| i | $\|C_i^d - C_i^{\text{native}}\|$ | $< 1$ | $< 2$ | $< 5$ | $< 10$ |
|---|---|---|---|---|---|
| 1 | 0.6 Å | x | x | x | x |
| 2 | 1.2 Å | | x | x | x |
| 3 | 1.9 Å | | x | x | x |
| 4 | 2.5 Å | | | x | x |
| 5 | 6.3 Å | | | | x |
| | | 20% | 60% | 80% | 100% |

**Local Distance Difference Test (LDDT)** Local Distance Difference Test (LDDT), is a residue-level score that does not require alignment of the structures and compares instead the local neighborhood of every residue, in the decoy and in the native structure. If we define the neighborhood of a residue as the set of its contacts, i.e. the set of other residues that lie within a certain distance from it, we can express the quality of that residue as the percentage of contacts that it shares with the corresponding residue in the native structure.

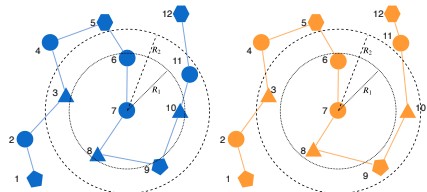

Figure 8: Example of LDDT scoring for residue 7: the residues within a radius $R_1$ are $\{6, 8, 10\}$ the native structure (left) and $\{6, 8\}$ for the decoy (right); at a radius $R_2$ we have $\{3, 6, 8, 9, 10, 11\}$ the native structure (left) and $\{3, 6, 8, 9, 10\}$ for the decoy (right).

# B DATASETS AND STATISTICS

## B.1 DATASETS

We consider all decoys of all target included in CASP 7-13, excluding proteins whose sequence is shorter than 50 residues and targets that have been canceled by the organizers. In addition to CASP datasets, we test our method on all targets published in CAMEO (Haas et al., 2013) between July and December 2017.

Table 3: Datasets from previous CASP editions

| Dataset | Targets | Models | Usage |
|---------|---------|--------|-------|
| CASP 7  | 95      | 19591  |           |
| CASP 8  | 122     | 34789  | Train/Val |
| CASP 9  | 117     | 34946  |           |
| CASP 10 | 103     | 26254  |           |
| CASP 11 | 83      | 16094  |           |
| CASP 12 | 40      | 6924   | Test      |
| CASP 13 | 20      | 7838   |           |
| CAMEO   | 676     | 20891  |           |

## B.2 PROTEIN GRAPHS STATISTICS

The cutoff value $d_{\max}$ determines which edges are included in the graph and, consequentially, its connectivity. A low cutoff implies a sparsely connected graph, with few edges and long paths between nodes. A higher cutoff yields a denser graph with more edges and shorter paths.

In figure 9 we report some statistics about number of edges, average degree and average shortest paths, evaluated at different cutoff values on 1700 decoys from CASP 11.

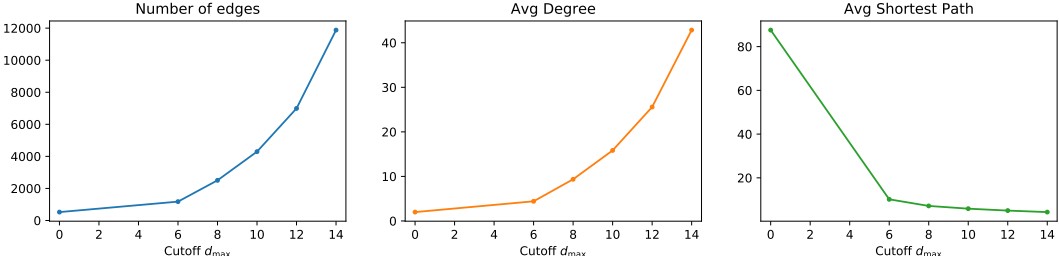

Figure 9: Number of edges, average degree, and average shortest paths at different cutoff values (sample size: 1700 decoys from CASP 11)

# C   GRAPHQA ARCHITECTURE

In this section, we illustrate in more detail the structure of the GRAPHQA architecture, as well as the hyperparameter space that was explored to optimize performances on the validation set.

## C.1   MESSAGE-PASSING LAYERS

Within GRAPHQA, a protein structure is represented as a graph whose nodes correspond to residues and whose edges connect interacting pairs of amino acids. At the input, the features of the $i$-th residue are encoded in a node feature vector $\boldsymbol{v}_i$. Similarly, the features of the pairwise interaction between residues $i$ and $j$ are encoded in an edge feature vector $\boldsymbol{e}_{i,j}$. A global bias term $\boldsymbol{u}$ is also added to represent information that is not localized to any specific node/edge of the graph.

With this graph representation, one layer of message passing performs the following updates.

1. For every edge $i \rightarrow j$, the edge feature vector is updated using a function $\phi^e$ of adjacent nodes $\boldsymbol{v}_i$ and $\boldsymbol{v}_j$, of the edge itself $\boldsymbol{e}_{i,j}$ and of the global attribute $\boldsymbol{u}$:

$$\boldsymbol{e}'_{i,j} = \phi^e\left(\boldsymbol{e}_{i,j}, \boldsymbol{v}_i, \boldsymbol{v}_j, \boldsymbol{u}\right)$$

2. For every node $i$, features from incident edges $\{\,\boldsymbol{e}'_{j,i}\,\}$ are aggregated using a pooling function $\rho^{e \rightarrow v}$:

$$\bar{\boldsymbol{e}}'_i = \rho^{e \rightarrow v}\left(\{\,\boldsymbol{e}'_{j,i}\,\}\right)$$

3. For every node $i$, the node feature vector is updated using a function $\phi^v$ of aggregated incident edges $\bar{\boldsymbol{e}}'_i$, of the node itself $\boldsymbol{v}_i$ and of the global attribute $\boldsymbol{u}$:

$$\boldsymbol{v}'_i = \phi^v\left(\bar{\boldsymbol{e}}'_i, \boldsymbol{v}_i, \boldsymbol{u}\right)$$

4. All edges are aggregated using a pooling function $\rho^{e \rightarrow u}$:

$$\bar{\boldsymbol{e}}' = \rho^{e \rightarrow u}\left(\{\,\boldsymbol{e}'_{i,j}\,\}\right)$$

5. All nodes are aggregated using a pooling function $\rho^{v \rightarrow u}$:

$$\bar{\boldsymbol{v}}' = \rho^{v \rightarrow u}\left(\{\,\boldsymbol{v}'_i\,\}\right)$$

6. The global feature vector is updated using a function $\phi^u$ of the aggregated edges $\bar{\boldsymbol{e}}'$, of the aggregated nodes $\bar{\boldsymbol{v}}'$ and of the global attribute $\boldsymbol{u}$:

$$\boldsymbol{u}' = \phi^u\left(\bar{\boldsymbol{e}}', \bar{\boldsymbol{v}}', \boldsymbol{u}\right)$$

In GRAPHQA, all intermediate updates are implemented as Linear-Dropout-ReLU functions, and all aggregation functions use average pooling. The encoder and readout layers do not make use of message passing, effectively processing every node/edge in isolation. Message passing is instead enabled for the core layers of the network and enables GRAPHQA to process information within progressively expanding neighborhoods.

The number of neurons in the core message-passing layers decreases from the input to the output. Specifically it follows a linear interpolation between the input and output numbers reported below, rounded to the closest power of two. In preliminary experiments, we noticed that a progressive increase of the number of layers results in convergence issues, which is in contrast to the practice of increasing the number of channels in Convolutional Neural Networks.

## C.2 HYPERPARAMETER OPTIMIZATION

We perform a guided grid search over the following hyper parameter space. The final model is chosen to be the one with the highest $R_{\text{target}}$ on the validation set. The following considerations were made:

- The values for $d_{\text{max}}$ are chosen on the base that the typical bond length is $\sim 5$Å and residue-residue interactions are negligible after $\sim 10$Å.

- The values for $\sigma$ are chosen so that the RBF encoding of the edge length is approximately linear around $\sim 5$Å.

- The values for $L$ are chosen to approximately match the average length of the shortest paths in the protein graphs at different cutoffs.

- In addition to what described in section 2.3, we also tested an architecture with BatchNorm layers between the Dropout and ReLU operations, but apart from a significant slowdown we did not notice any improvement.

Table 4: Hyper parameter space and best values

| Hyper parameter | Values | Best |
|---|---|---|
| MP Layers $L$ | 3, 4, 5, 6, 7, 8, 9 | 6 |
| MP input size $e$ | 32, 64, 128 | 128 |
| MP input size $v$ | 64, 128, 256, 512 | 512 |
| MP input size $u$ | 64, 128, 256, 512 | 512 |
| MP output size $e$ | 8, 12, 16, 32 | 16 |
| MP output size $v$ | 8, 16, 32, 64 | 64 |
| MP output size $u$ | 8, 12, 16, 32 | 32 |
| Cutoff $d_{\text{max}}$ | 6, 8, 10, 12 | 8 |
| Sigma $\sigma$ | 10, 15, 20 | 15 |
| Dropout rate | 0, 0.1 0.2, 0.3, 0.4 | 0.2 |
| Learning rate | $10^{-2}, 10^{-3}$ | $10^{-3}$ |
| Weight decay | $10^{-4}, 10^{-5}$ | $10^{-5}$ |
| Local weight $\lambda^{\ell}$ | 1, 5, 10 | 1 |
| Global weight $\lambda^{g}$ | 1, 5, 10 | 1 |

# D    ADDITIONAL STUDIES

To complement the analysis reported in the main text, we perform additional studies on the effect of feature representation and on the generalization ability of the trained model.

## D.1    DISTANCE AND SEPARATION ENCODING

The feature vectors associated to the edge of the graph represent two types of distances between residues, namely spatial distance and separation in the sequence. In this study we evaluate the effect of different representations on validation performances.

Spatial distance is the physical distance between ammino acids, measured as the euclidean distance between their $\beta$ carbon atoms. We consider three possible encodings for this distance:

- **Absent:** spatial distance is not provided as input;
- **Scalar:** spatial distance is provided as a raw scalar value (in Ångstrom);
- **RBF:** spatial distance is encoded using 32 RBF kernels, with unit variance, equally spaced between 0 and 20.

Figure 10 reports the aggregated performances on CASP 11 of ten runs for each of the above. The rich representation of the RBF kernels seem to improve both LDDT and GDT_TS scoring performances, even though the effect is rather limited.

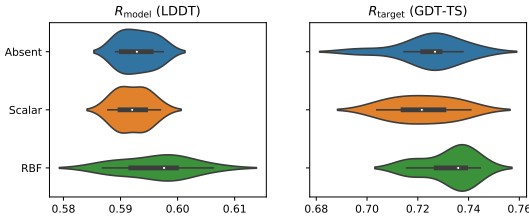

Figure 10: Spatial distance: absent, encoded as a scalar, encoded using RBF kernels.

Separation is the number of residues between to amino acids in the sequence, we consider three possible encodings:

- **Absent:** sequential separation is not provided as input;
- **Scalar:** sequential separation is provided as a raw scalar value (positive integer);
- **Categorical:** sequential separation is encoded as a one-hot categorical variable, according to the classes $\{\,0, 1, 2, 3, 4, 5 : 10, > 10\,\}$, which are based on typical interaction patterns within a peptidic chain.

Figure 11 reports the aggregated performances on CASP 11 of ten runs for each of the above. For local scoring, the choice of encoding plays little difference as long as separation is present in the input. On the other hand, the choice of categorical encoding over scalar encoding results in higher global scoring performance.

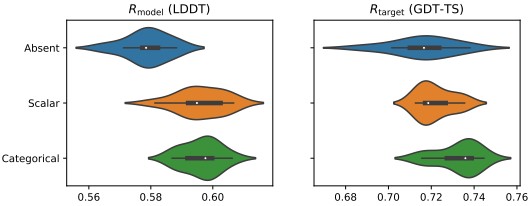

Figure 11: Sequential separation: absent, encoded as a scalar, encoded as a categorical variable.

## D.2 Transmembrane vs. soluble proteins

In this study, we evaluate how the natural environment of a protein affects the predictive performances of our method. Targets from CASP 11 and 12 are classified as transmembrane and soluble according to Peters et al. (2015) and scored separately using GRAPHQA. Transmembrane proteins behave differently from soluble proteins as a consequence of the environment they are placed in. The former expose non-polar residues to the cellular membrane that surrounds their structure. On the contrary, the latter tend to present polar amino acids to the surrounding water-based solvent.

Since this information is not explicitly provided to the model, we can compare the predictive performances between the two sets and check that it has actually learned a flexible protein representation. The outcome of this evaluation is shown in table 5. While it is evident that GRAPHQA performs better on soluble proteins, which are more numerous in the training set, it also scores transmembrane proteins to an acceptable degree.

Table 5: Performances of GRAPHQA on transmembrane and soluble targets from CASP 11 and 12.

| | GDT_TS | | | | | | | | LDDT | | |
| | FRL | $R$ | $R_{target}$ | RMSE | $\rho$ | $\rho_{target}$ | $\tau$ | $\tau_{target}$ | $R$ | $R_{model}$ | RMSE |
|---|---|---|---|---|---|---|---|---|---|---|---|
| Transmembrane | 0.092 | 0.714 | 0.538 | 0.209 | 0.714 | 0.463 | 0.523 | 0.328 | 0.679 | 0.487 | 0.170 |
| Soluble | 0.076 | 0.790 | 0.627 | 0.177 | 0.796 | 0.500 | 0.594 | 0.361 | 0.770 | 0.532 | 0.155 |

# E    ADDITIONAL METRICS

The Quality Assessment literature is rich of metrics to measure the performances of a scoring method. In the main text we tried to keep the exposition uncluttered by only reporting figures for the most important metrics. Here we present a more extensive set of metrics, that further describe our method and can serve as future benchmark.

In the following, we use:

$$
\begin{aligned}
&t = 1, \ldots, T && \text{Target proteins}\\
&d = 1, \ldots, D^t && \text{Decoys of a target}\\
&i, j = 1, \ldots, |S^t| && \text{Residue indexes of a target}\\
&q^{g,t,d} = \text{GDT\_TS}(C^{t,d}, C^{t,\text{native}}) && \text{Global quality score (true)}\\
&q_i^{\ell,t,d} = \text{LDDT}(C^{t,d}, C^{t,\text{native}}) && \text{Local quality scores (true)}\\
&\text{GRAPHQA}^g(\mathcal{P}^{t,d}) && \text{Global quality score (predicted)}\\
&\text{GRAPHQA}_i^{\ell}(\mathcal{P}^{t,d}) && \text{Local quality scores (predicted)}
\end{aligned}
$$

**Root Mean Squared Error (RMSE)** We compute RMSE between all true and predicted scores, for both LDDT and GDT_TS.

For LDDT, it it the square root of:

$$
\text{MSE} = \frac{1}{T} \sum_{t=1}^{T} \frac{1}{D^t} \sum_{d=1}^{D^t} \frac{1}{|S^t|} \sum_{i=1}^{|S^t|} \left( q_i^{\ell,t,d} - \text{GRAPHQA}_i^{\ell}(\mathcal{P}^{t,d}) \right)^2
$$

For GDT_TS, it it the square root of:

$$
\text{MSE} = \frac{1}{T} \sum_{t=1}^{T} \frac{1}{D^t} \sum_{d=1}^{D^t} \left( q^{g,t,d} - \text{GRAPHQA}^g(\mathcal{P}^{t,d}) \right)^2
$$

**Correlation coefficients** We compute the Pearson ($R$), Spearman ($\rho$) and Kendall ($\tau$) correlation coefficients between *all* true and predicted scores. Since all scores are treated equally, with no distinction between different decoys or different targets, a high value for these scores can be misleading. Thus, their *per-model* and *per-target* versions should be also checked.

For LDDT:

$$
R = \text{PEARSON}\left( \{\, q_i^{\ell,t,d} \,\}, \{\, \text{GRAPHQA}_i^{\ell}(\mathcal{P}^{t,d}) \,\} \right)
$$

$$
\rho = \text{SPEARMAN}\left( \{\, q_i^{\ell,t,d} \,\}, \{\, \text{GRAPHQA}_i^{\ell}(\mathcal{P}^{t,d}) \,\} \right)
$$

$$
\tau = \text{KENDALL}\left( \{\, q_i^{\ell,t,d} \,\}, \{\, \text{GRAPHQA}_i^{\ell}(\mathcal{P}^{t,d}) \,\} \right)
$$

For GDT_TS:

$$
R = \text{PEARSON}\left( \{\, q^{g,t,d} \,\}, \{\, \text{GRAPHQA}^g(\mathcal{P}^{t,d}) \,\} \right)
$$

$$
\rho = \text{SPEARMAN}\left( \{\, q^{g,t,d} \,\}, \{\, \text{GRAPHQA}^g(\mathcal{P}^{t,d}) \,\} \right)
$$

$$
\tau = \text{KENDALL}\left( \{\, q^{g,t,d} \,\}, \{\, \text{GRAPHQA}^g(\mathcal{P}^{t,d}) \,\} \right)
$$

**Correlation coefficients per-model** For every decoy of every target, we compute the Pearson ($R_{\text{model}}$), Spearman ($\rho_{\text{model}}$) and Kendall ($\tau_{\text{model}}$) correlation coefficients between true and predicted residue-level scores (LDDT). We then report the average correlation coefficients across all decoys of all targets. The *per-model* correlation coefficients estimate the performance of the network to rank individual residues by their quality and distinguish correctly vs. incorrectly folded segments.

Per-model correlation coefficients are computed only for LDDT:

$$R_{\text{model}} = \frac{1}{T} \sum_{t=1}^{T} \frac{1}{D^t} \sum_{d=1}^{D^t} \text{PEARSON}\left(\left\{ q_i^{\ell,t,d} \right\}, \left\{ \text{GRAPHQA}_i^{\ell}(\mathcal{P}^{t,d}) \right\}\right)$$

$$\rho_{\text{model}} = \frac{1}{T} \sum_{t=1}^{T} \frac{1}{D^t} \sum_{d=1}^{D^t} \text{SPEARMAN}\left(\left\{ q_i^{\ell,t,d} \right\}, \left\{ \text{GRAPHQA}_i^{\ell}(\mathcal{P}^{t,d}) \right\}\right)$$

$$\tau_{\text{model}} = \frac{1}{T} \sum_{t=1}^{T} \frac{1}{D^t} \sum_{d=1}^{D^t} \text{KENDALL}\left(\left\{ q_i^{\ell,t,d} \right\}, \left\{ \text{GRAPHQA}_i^{\ell}(\mathcal{P}^{t,d}) \right\}\right)$$

**Correlation coefficients per-target** For every target, we compute the Pearson ($R_{\text{target}}$), Spearman ($\rho_{\text{target}}$) and Kendall ($\tau_{\text{target}}$) correlation coefficients between true and predicted decoy-level scores (GDT_TS). We then report the average correlation coefficients across all targets. With reference to the funnel plots, this would be the correlation between the markers in every plot, averaged across all plots. The *per-target* correlation coefficients estimate the performance of the network to rank the decoys of a target by their quality and select the ones with highest global quality.

Per-target correlation coefficients are computed only for GDT_TS:

$$R_{\text{target}} = \frac{1}{T} \sum_{t=1}^{T} \text{PEARSON}\left(\left\{ q^{g,t,d} \right\}, \left\{ \text{GRAPHQA}^g(\mathcal{P}^{t,d}) \right\}\right)$$

$$\rho_{\text{target}} = \frac{1}{T} \sum_{t=1}^{T} \text{SPEARMAN}\left(\left\{ q^{g,t,d} \right\}, \left\{ \text{GRAPHQA}^g(\mathcal{P}^{t,d}) \right\}\right)$$

$$\tau_{\text{target}} = \frac{1}{T} \sum_{t=1}^{T} \text{KENDALL}\left(\left\{ q^{g,t,d} \right\}, \left\{ \text{GRAPHQA}^g(\mathcal{P}^{t,d}) \right\}\right)$$

**First Rank Loss (FRL)** For every target, we compute the difference in GDT_TS between the best decoy according to ground-truth scores and best decoy according to the predicted scores. We then report the average FRL across all targets. This represents the loss in (true) quality we would suffer if we were to choose a decoy according to our rankings and can is represented in the funnel plots by the gap between the two vertical lines indicating the true best (green) and predicted best (red).

FRL measures the ability to select a single best decoy for a given target. In our experiments, however, we noticed that FRL is extremely subject to noise, as it only considers top-1 decoys. Therefore, we consider NDCG to be a superior metric for this purpose, though we have not seen it used in the QA literature.

FRL is only computed for GDT_TS:

$$\text{FRL} = \frac{1}{T} \sum_{t=1}^{T} \left| \max\left\{ q^{g,t,d} \right\} - q^{g,t,d*} \right|,$$

where $d* = \arg\max_d \left\{ \text{GRAPHQA}^g(\mathcal{P}^{t,d}) \right\}$ for every target $t$.

**Recall at $k$ (REC@$k$)** We can describe Quality Assessment as an information retrieval task, where every target represents a query and its decoys are the documents available for retrieval. If we consider the best decoy to have a score of 1 and all others to have zero score, we can compute the average REC@$k$ as the percentage of queries for which the best decoy is retrieved among the top-$k$ results.

This metric, however, is subject to the same pitfalls of FRL, since it only considers the best decoy of every target and ignores the relevance of the others. As described below, NDCG offers a better perspective over the decoys retrieved by a QA method.

**Normalized Discounted Cumulative Gain at $k$ (NDCG@$k$)** For a given query we consider the top-$k$ decoys ranked according to their predicted global scores. Discounted Cumulative Gain at $k$

(DCG@$k$) is computed as the cumulative sum of their ground-truth GDT_TS scores (gain), discounted however according to the position in the list. A high DCG@$k$ is obtained therefore by a) selecting the $k$-best decoys to be part of the top-$k$ predictions, and b) sorting them in order of decreasing quality (the higher in the list, the lower the discount).

Dividing DCG@$k$ by DCG$^{\text{ideal}}$@$k$ (obtained by ranking according to the ground-truth scores), yields the Normalized Discounted Cumulative Gain NDCG@$k \in [0, 1]$, which can be compared and averaged across targets.

## F  ADDITIONAL RESULTS

In this section we present additional results for datasets, dataset splits, methods, and metrics that are excluded from the main text for sake of brevity. The datasets considered in the following pages are: CASP 11, CASP 12, CASP 13, CAMEO.

The CASP 11 and 12 datasets are conventionally divided into: *stage 1*, containing 20 randomly-selected decoys per target, and *stage 2*, containing the top-150 decoys of each target. In the QA literature, some papers report results either on the dataset as a whole, or on the stage 1 and stage 2 splits. Furthermore, some papers report performances of other methods that differ from the original papers for reasons that are left unspecified. In the main text, we adhere to the following rules to summarize the metrics we collected:

- Metrics computed on stage 1 are considered noisy and ignored, since stage 1 splits contain only 20 randomly-selected decoys per target
- Metrics computed on stage 2 and on the whole dataset are considered equally valid, allowing to "merge" results from papers with different scoring strategies
- If multiple values are reported from multiple sources for the same (method, dataset) pair, only the best one is reported

## F.1 CASP 11

| TestSet | Method | Source | GDT_TS | | | | | | | | LDDT | | |
|---|---|---|---|---|---|---|---|---|---|---|---|---|---|
| | | | FRL | R | $R_{target}$ | RMSE | $\rho$ | $\rho_{target}$ | $\tau$ | $\tau_{target}$ | R | $R_{model}$ | RMSE |
| CASP 11 | ProQ3D | ProQ4 | | | | | | | | | 0.84 | 0.61 | 0.125 |
| | ProQ4 | ProQ4 | | | | | | | | | 0.77 | 0.56 | 0.147 |
| | GRAPHQA$_{RAW}$ | Ours | 0.082 | 0.836 | 0.609 | 0.146 | 0.837 | 0.49 | 0.637 | 0.354 | 0.799 | 0.516 | 0.14 |
| | GRAPHQA | Ours | 0.071 | 0.91 | 0.74 | 0.117 | 0.918 | 0.622 | 0.747 | 0.461 | 0.852 | 0.602 | 0.122 |
| stage 1 | ProQ3D | 3D CNN | 0.046 | | 0.755 | | | 0.673 | | 0.529 | | | |
| | | Ornate | 0.066 | 0.795 | 0.691 | | 0.782 | 0.606 | 0.58 | 0.462 | | | |
| | VoroMQA | 3D CNN | 0.087 | | 0.637 | | | 0.521 | | 0.394 | | | |
| | | Ornate | 0.085 | 0.689 | 0.617 | | 0.682 | 0.482 | 0.483 | 0.361 | | | |
| | RWplus | 3D CNN | 0.122 | | 0.512 | | | 0.402 | | 0.303 | | | |
| | | Ornate | 0.128 | 0.08 | 0.467 | | 0.003 | 0.371 | -0.016 | 0.274 | | | |
| | 3D CNN | 3D CNN | 0.064 | | 0.535 | | | 0.425 | | 0.325 | | | |
| | | Ornate | 0.104 | 0.532 | 0.442 | | 0.614 | 0.369 | 0.437 | 0.28 | | | |
| | Ornate | Ornate | 0.077 | 0.635 | 0.465 | | 0.634 | 0.372 | 0.44 | 0.275 | | | |
| | GRAPHQA$_{RAW}$ | Ours | 0.09 | 0.829 | 0.63 | 0.135 | 0.81 | 0.514 | 0.609 | 0.393 | 0.79 | 0.475 | 0.131 |
| | GRAPHQA | Ours | 0.035 | 0.923 | 0.788 | 0.09 | 0.924 | 0.647 | 0.755 | 0.515 | 0.861 | 0.57 | 0.108 |
| stage 2 | ProQ3D | 3D CNN | 0.066 | | 0.452 | | | 0.433 | | 0.307 | | | |
| | | Ornate | 0.053 | 0.772 | 0.444 | | 0.796 | 0.432 | 0.594 | 0.304 | | | |
| | VoroMQA | 3D CNN | 0.063 | | 0.457 | | | 0.499 | | 0.321 | | | |
| | | Ornate | 0.066 | 0.651 | 0.419 | | 0.688 | 0.412 | 0.505 | 0.291 | | | |
| | RWplus | 3D CNN | 0.089 | | 0.206 | | | 0.248 | | 0.176 | | | |
| | | Ornate | 0.088 | 0.056 | 0.167 | | 0.033 | 0.192 | 0.011 | 0.137 | | | |
| | 3D CNN | 3D CNN | 0.064 | | 0.421 | | | 0.409 | | 0.288 | | | |
| | | Ornate | 0.074 | 0.629 | 0.375 | | 0.655 | 0.363 | 0.433 | 0.254 | | | |
| | Ornate | Ornate | 0.055 | 0.637 | 0.386 | | 0.673 | 0.371 | 0.475 | 0.259 | | | |
| | GRAPHQA$_{RAW}$ | Ours | 0.071 | 0.82 | 0.379 | 0.149 | 0.82 | 0.357 | 0.618 | 0.251 | 0.787 | 0.529 | 0.142 |
| | GRAPHQA | Ours | 0.063 | 0.899 | 0.539 | 0.123 | 0.905 | 0.507 | 0.729 | 0.363 | 0.839 | 0.61 | 0.126 |

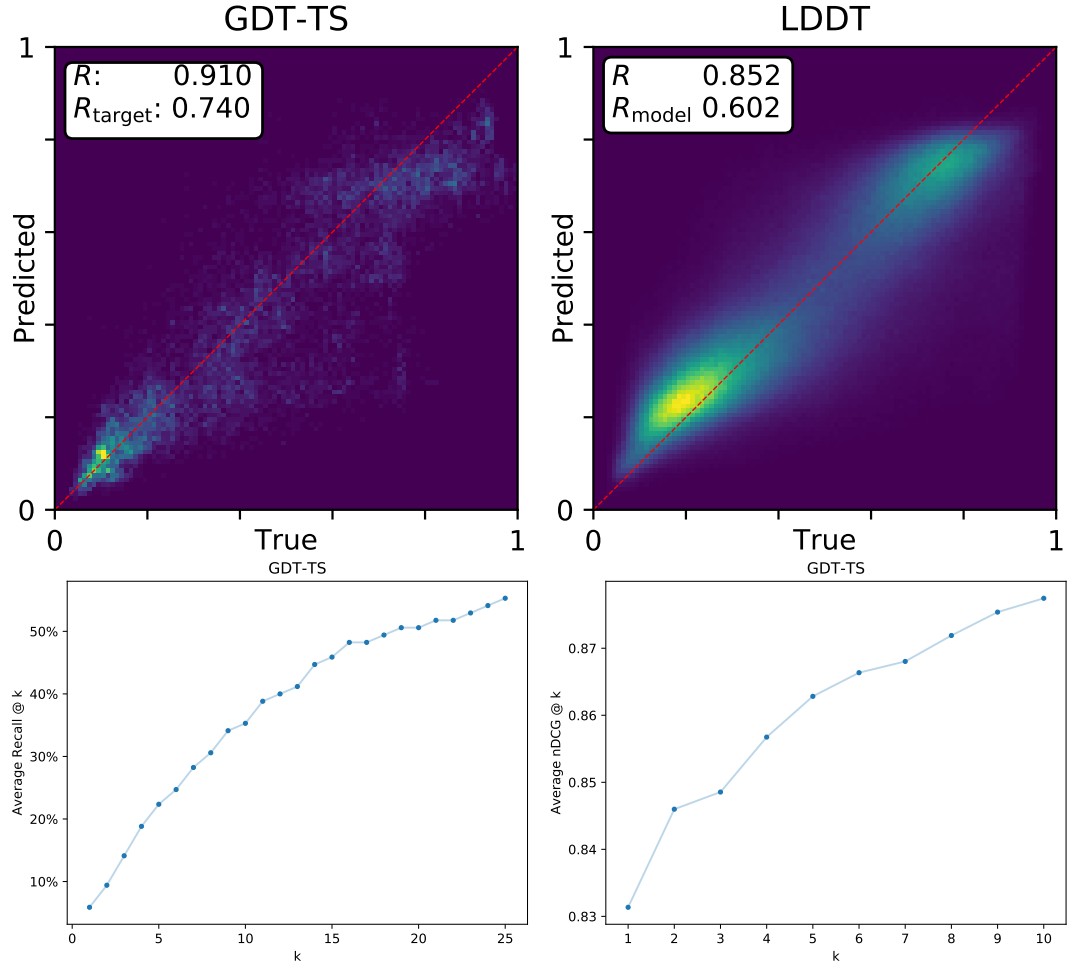

Figure 12: CASP 11: Histograms of true vs. predicted LDDT and GDT_TS scores, average recall @ k, average NDCG @ k

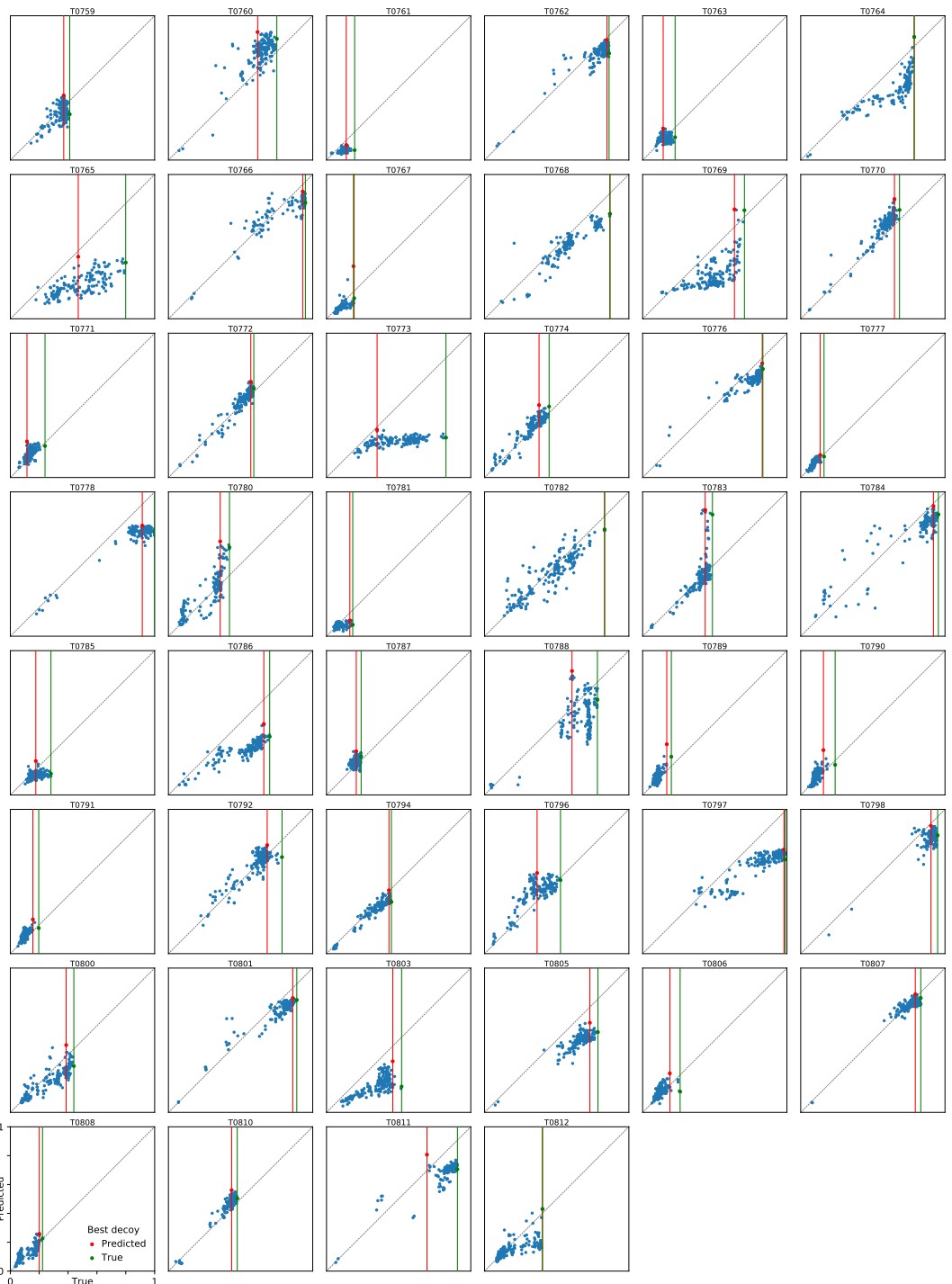

Figure 13: CASP 11: funnels (continues)

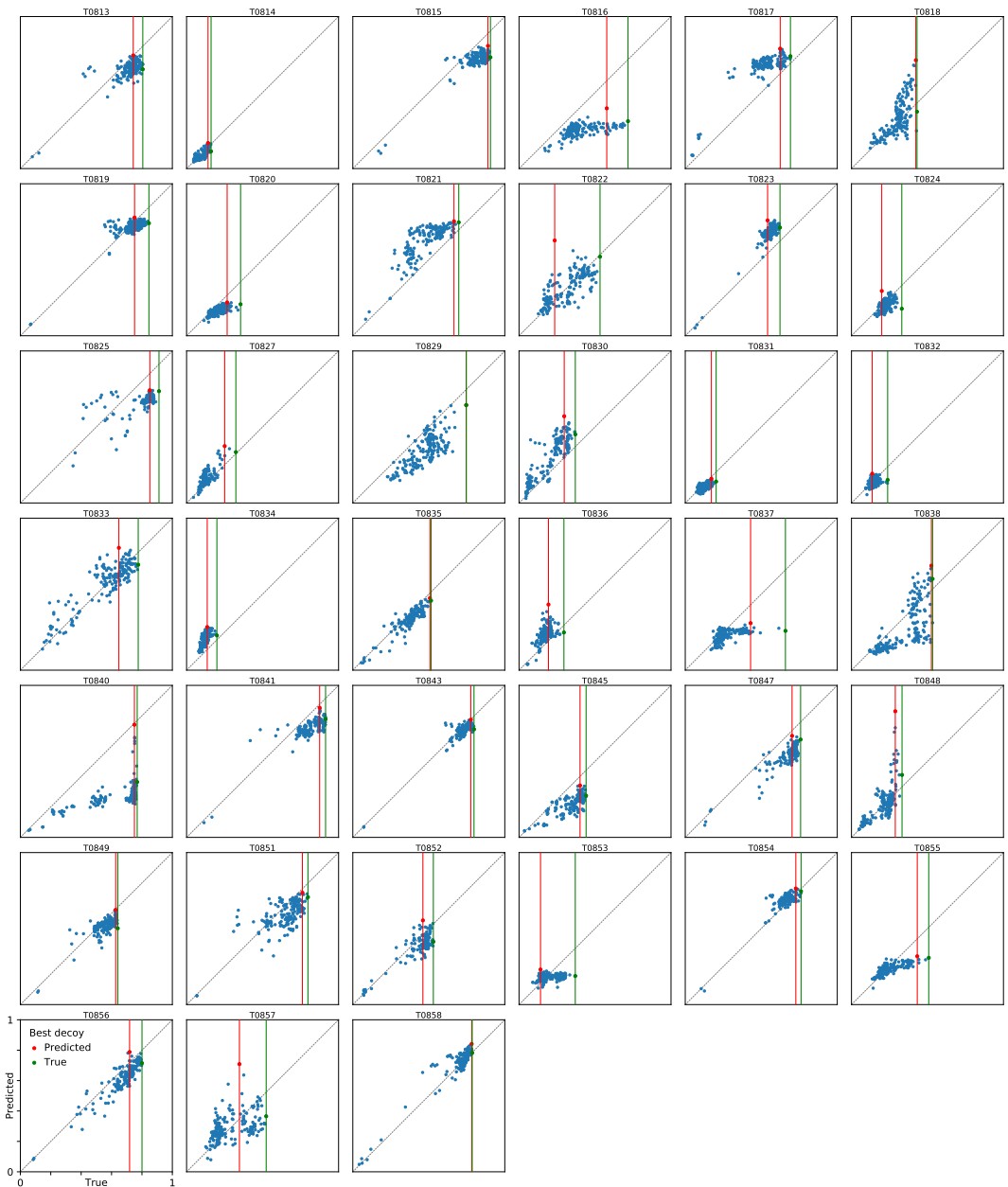

Figure 14: CASP 11: funnels (continued)

## F.2 CASP 12

| TestSet | Method | Source | GDT_TS | | | | | | | | LDDT | | | | |
|---|---|---|---|---|---|---|---|---|---|---|---|---|---|---|---|
| | | | FRL | R | $R_{target}$ | RMSE | $\rho$ | $\rho_{target}$ | $\tau$ | $\tau_{target}$ | R | $R_{model}$ | RMSE | $\rho$ | $\rho_{model}$ |
| CASP 12 | ProQ3D | 3D CNN | 0.164 | | 0.609 | | | 0.602 | | 0.451 | | | | | |
| | ProQ4 | Ours | | | | | | | | | 0.772 | 0.516 | | 0.776 | 0.498 |
| | VoroMQA | 3D CNN | 0.161 | | 0.557 | | | 0.515 | | 0.38 | | | | | |
| | RWplus | 3D CNN | 0.192 | | 0.313 | | | 0.355 | | 0.257 | | | | | |
| | 3D CNN | 3D CNN | 0.146 | | 0.607 | | | 0.521 | | 0.381 | | | | | |
| | AngularQA | AngularQA | 0.138 | 0.651 | 0.439 | | | | | | | | | | |
| | GRAPHQA$_{RAW}$ | Ours | 0.092 | 0.816 | 0.673 | 0.149 | 0.814 | 0.606 | 0.624 | 0.448 | 0.796 | 0.501 | 0.139 | | |
| | GRAPHQA | Ours | 0.089 | 0.843 | 0.745 | 0.137 | 0.834 | 0.66 | 0.684 | 0.503 | 0.843 | 0.565 | 0.124 | | |
| stage 1 | ProQ3D | Ornate | 0.086 | 0.671 | 0.705 | | 0.478 | 0.636 | 0.335 | 0.482 | | | | | |
| | VoroMQA | Ornate | 0.085 | 0.456 | 0.611 | | 0.381 | 0.554 | 0.263 | 0.414 | | | | | |
| | RWplus | Ornate | 0.132 | -0.272 | 0.479 | | -0.538 | 0.465 | -0.381 | 0.344 | | | | | |
| | Ornate | Ornate | 0.113 | 0.551 | 0.566 | | 0.484 | 0.504 | 0.339 | 0.374 | | | | | |
| | AngularQA | AngularQA | 0.148 | | 0.502 | | | | | | | | | | |
| | GRAPHQA$_{RAW}$ | Ours | 0.068 | 0.721 | 0.679 | 0.127 | 0.623 | 0.596 | 0.451 | 0.448 | 0.661 | 0.413 | 0.118 | | |
| | GRAPHQA | Ours | 0.043 | 0.814 | 0.789 | 0.085 | 0.755 | 0.684 | 0.589 | 0.541 | 0.718 | 0.474 | 0.105 | | |
| stage 2 | ProQ3D | Ornate | 0.06 | 0.806 | 0.6 | | 0.8 | 0.54 | 0.601 | 0.388 | | | | | |
| | VoroMQA | Ornate | 0.106 | 0.605 | 0.559 | | 0.604 | 0.501 | 0.445 | 0.362 | | | | | |
| | RWplus | Ornate | 0.103 | -0.096 | 0.417 | | -0.096 | 0.378 | -0.067 | 0.265 | | | | | |
| | Ornate | Ornate | 0.072 | 0.67 | 0.491 | | 0.657 | 0.458 | 0.472 | 0.322 | | | | | |
| | AngularQA | AngularQA | 0.128 | | 0.377 | | | | | | | | | | |
| | GRAPHQA$_{RAW}$ | Ours | 0.094 | 0.807 | 0.614 | 0.151 | 0.807 | 0.545 | 0.618 | 0.395 | 0.793 | 0.507 | 0.141 | | |
| | GRAPHQA | Ours | 0.08 | 0.832 | 0.707 | 0.141 | 0.828 | 0.61 | 0.679 | 0.456 | 0.842 | 0.573 | 0.125 | | |

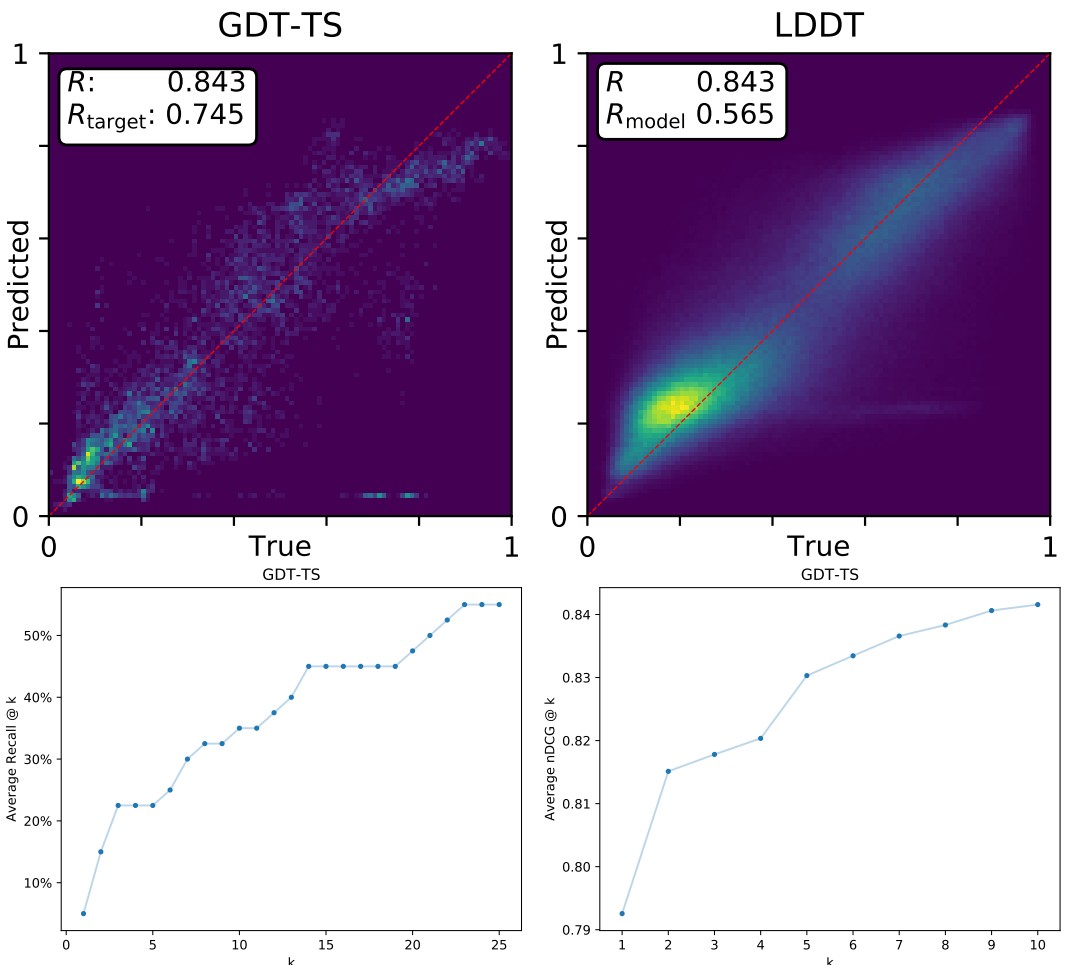

Figure 15: CASP 12: Histograms of true vs. predicted LDDT and GDT_TS scores, average recall @ k, average NDCG @ k

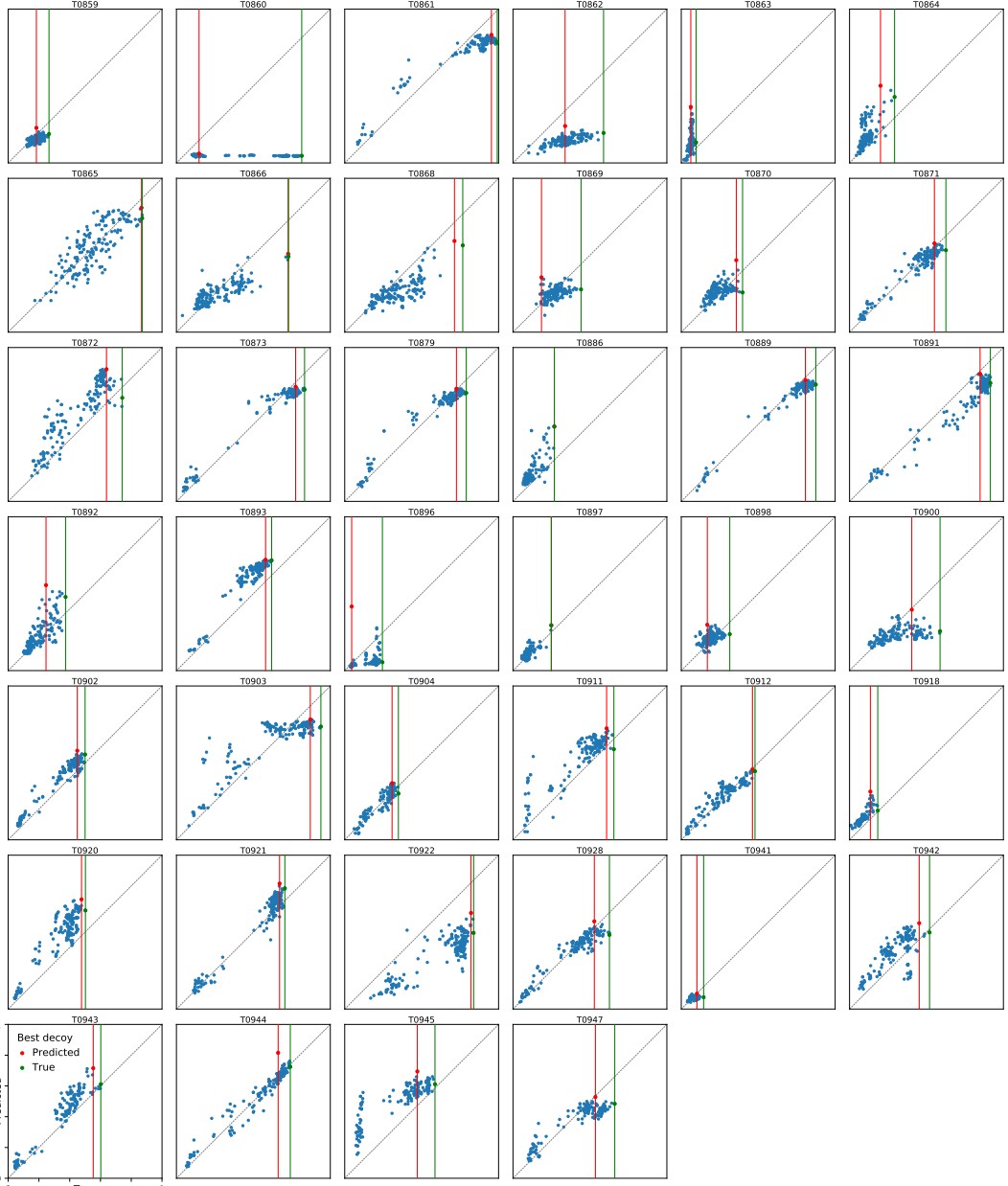

Figure 16: CASP 12: funnels

At a close analysis, it appears that the model completely fails to score decoys of target T060 and defaults to predicting a small constant value. To help pinpointing the problem, we compare the predictions made by GRAPHQA and GRAPHQA$_{\text{RAW}}$ (available in the repository). It turns out that the model trained on amino acid identity only does not output the same degenerate predictions as its fully-featured counterpart (the predictions are not perfect, but definitely better than a constant). We suspect that an error in the preprocessing pipeline might have produced misleading features for T060, e.g. the multiple sequence alignment program that extracts self information and partial entropy, or the DSSP program that computes secondary structure features.

## F.3 CASP 13

CASP 13 is the most recent edition of CASP and, at the moment, only few targets are available for public evaluation, i.e. their decoy structures are fully characterized, submitted predictions and ground-truth GDT_TS scores can be downloaded from the website. Here we present an evaluation on publicly available targets, while waiting to update these results as more data is released.

Also, it is important to note that GRAPHQA is only trained on CASP 7-10, while other participants have likely (re)trained their models on all previous CASP datasets as well as other datasets. However, even without retraining, we achieve performances that are in line with the results presented for CASP 11 and 12.

| TestSet | Method | Source | GDT_TS | | | | | | | | LDDT | | |
| | | | FRL | R | $R_{target}$ | RMSE | $\rho$ | $\rho_{target}$ | $\tau$ | $\tau_{target}$ | R | $R_{model}$ | RMSE |
|---|---|---|---|---|---|---|---|---|---|---|---|---|---|
| | ProQ3D | Ours | 0.160 | 0.849 | 0.671 | 0.129 | 0.810 | 0.619 | 0.625 | 0.458 | | | |
| | ProQ4 | Ours | 0.147 | 0.671 | 0.733 | 0.182 | 0.645 | 0.668 | 0.492 | 0.508 | | | |
| | VoroMQA | Ours | 0.024 | 0.767 | 0.665 | 0.177 | 0.765 | 0.606 | 0.571 | 0.443 | | | |
| CASP 13 | 3D CNN | Ours | 0.135 | 0.661 | 0.753 | 0.192 | 0.632 | 0.662 | 0.457 | 0.487 | | | |
| | Ornate | Ours | 0.283 | 0.533 | 0.646 | 0.352 | 0.522 | 0.642 | 0.360 | 0.467 | | | |
| | GRAPHQA$_{RAW}$ | Ours | 0.122 | 0.767 | 0.727 | 0.167 | 0.774 | 0.701 | 0.573 | 0.524 | 0.723 | 0.475 | 0.154 |
| | GRAPHQA | Ours | 0.093 | 0.846 | 0.834 | 0.123 | 0.840 | 0.799 | 0.647 | 0.616 | 0.764 | 0.539 | 0.142 |

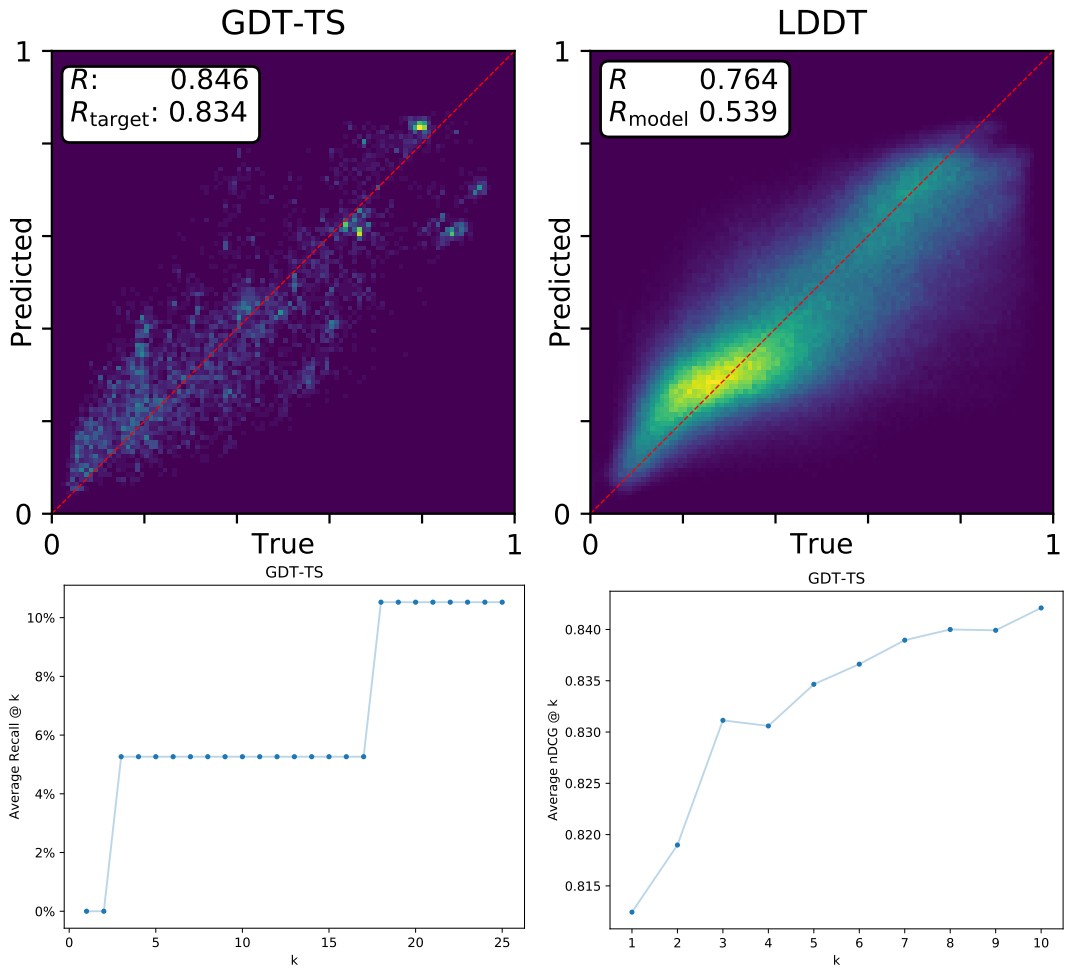

Figure 17: CASP 13: Histograms of true vs. predicted LDDT and GDT_TS scores, average recall @ k, average NDCG @ k

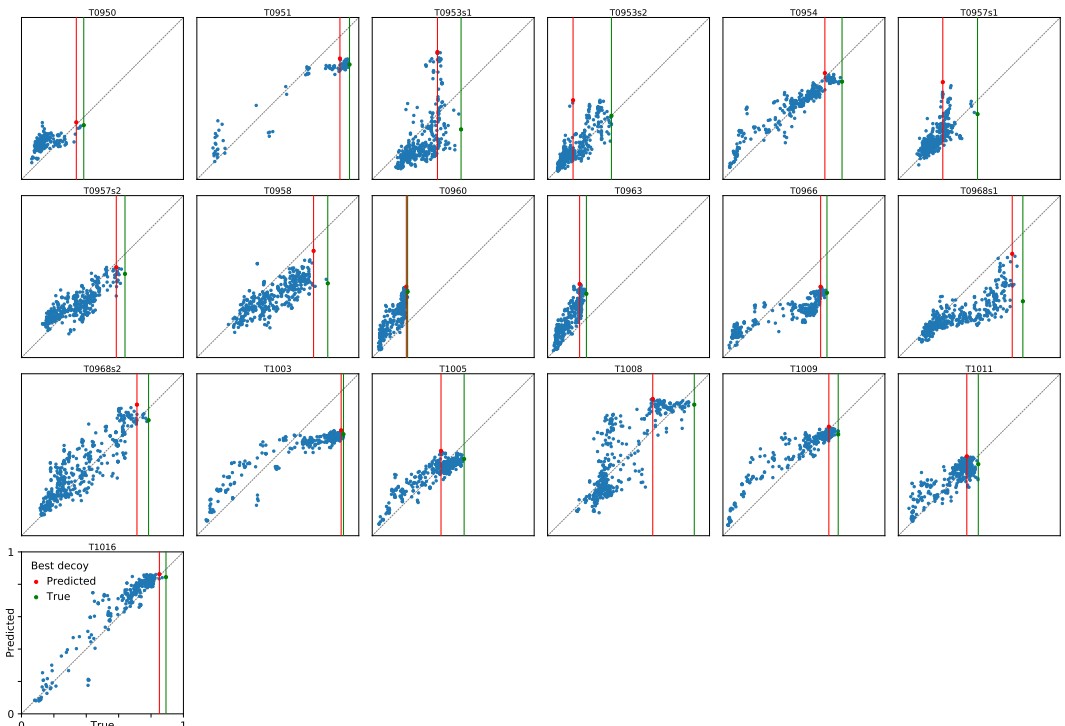

Figure 18: CASP 13: funnels

## F.4 CAMEO

| TestSet | Method | Source | GDT_TS | | | | | | | | LDDT | | |
|---|---|---|---|---|---|---|---|---|---|---|---|---|---|
| | | | FRL | R | $R_{target}$ | RMSE | $\rho$ | $\rho_{target}$ | $\tau$ | $\tau_{target}$ | R | $R_{model}$ | RMSE |
| | ProQ3D | ProQ4 | | | | | | | | | 0.79 | 0.64 | 0.137 |
| | ProQ4 | ProQ4 | | | | | | | | | 0.65 | 0.56 | 0.201 |
| | VoroMQA | 3D CNN | 0.099 | | 0.456 | | | 0.427 | | 0.346 | | | |
| CAMEO | RWplus | 3D CNN | 0.162 | | 0.122 | | | 0.095 | | 0.068 | | | |
| | 3D CNN | 3D CNN | 0.06 | | 0.586 | | | 0.532 | | 0.426 | | | |
| | GRAPHQA$_{RAW}$ | Ours | 0.046 | 0.725 | 0.585 | 0.17 | 0.675 | 0.505 | 0.497 | 0.379 | 0.699 | 0.56 | 0.169 |
| | GRAPHQA | Ours | 0.044 | 0.747 | 0.598 | 0.197 | 0.666 | 0.525 | 0.495 | 0.399 | 0.748 | 0.616 | 0.161 |

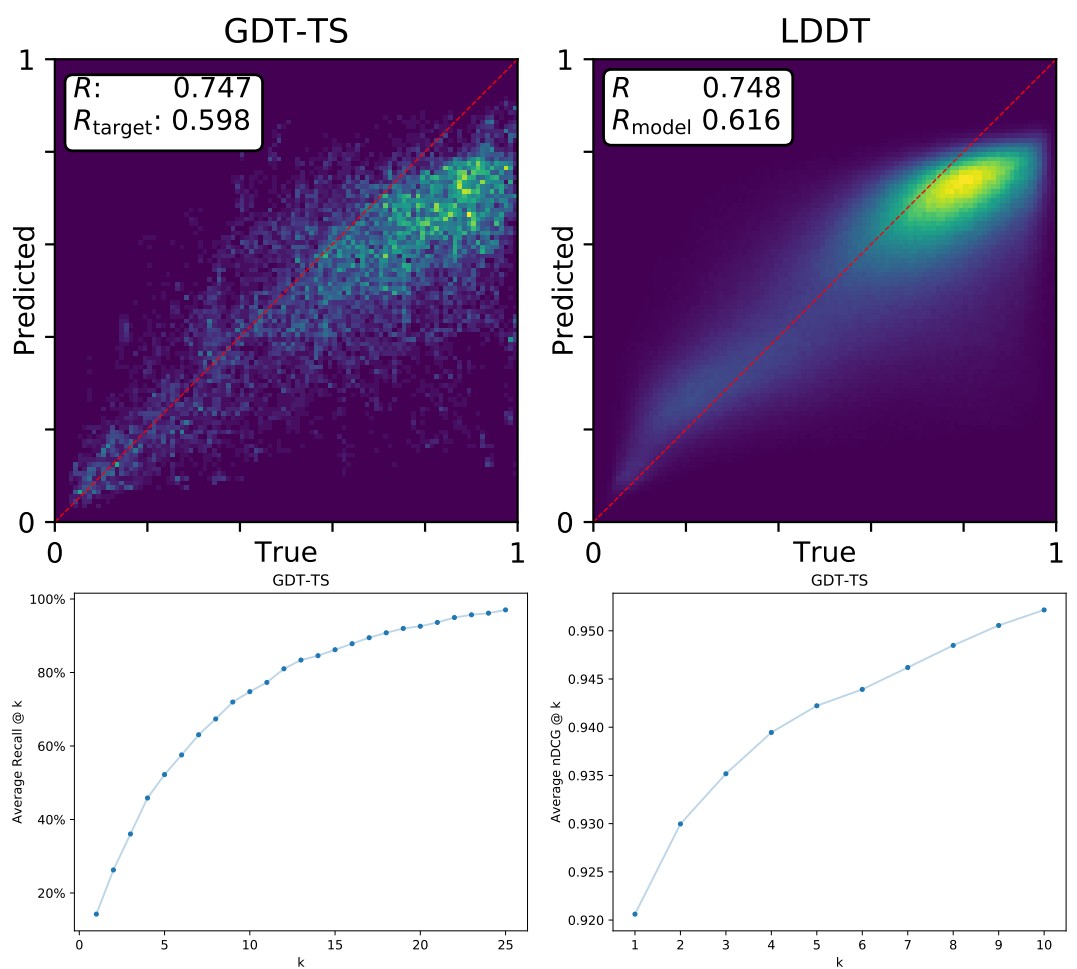

Figure 19: CAMEO: Histograms of true vs. predicted LDDT and GDT_TS scores, average recall @ k, average NDCG @ k

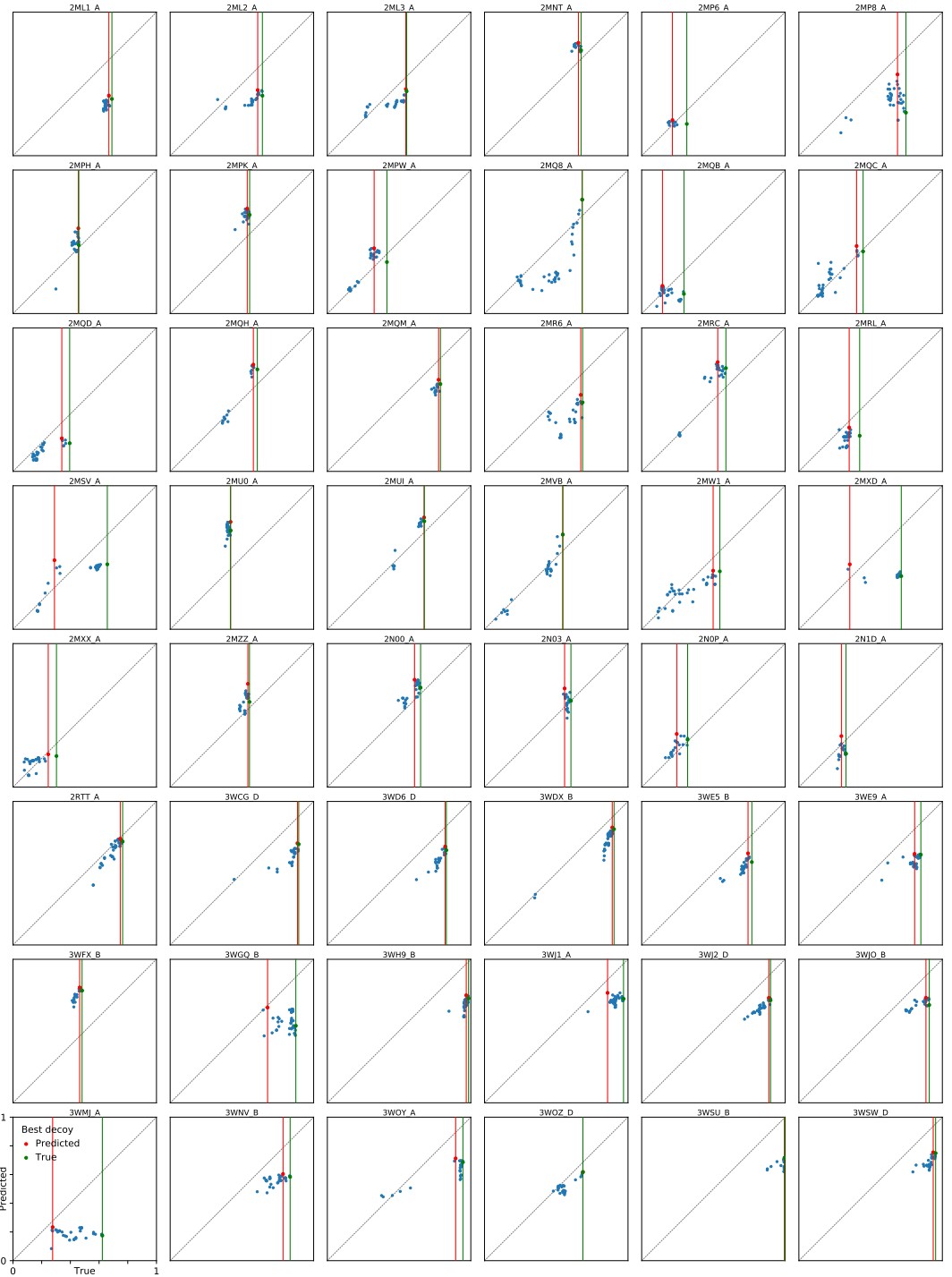

Figure 20: CAMEO: funnels (continues)

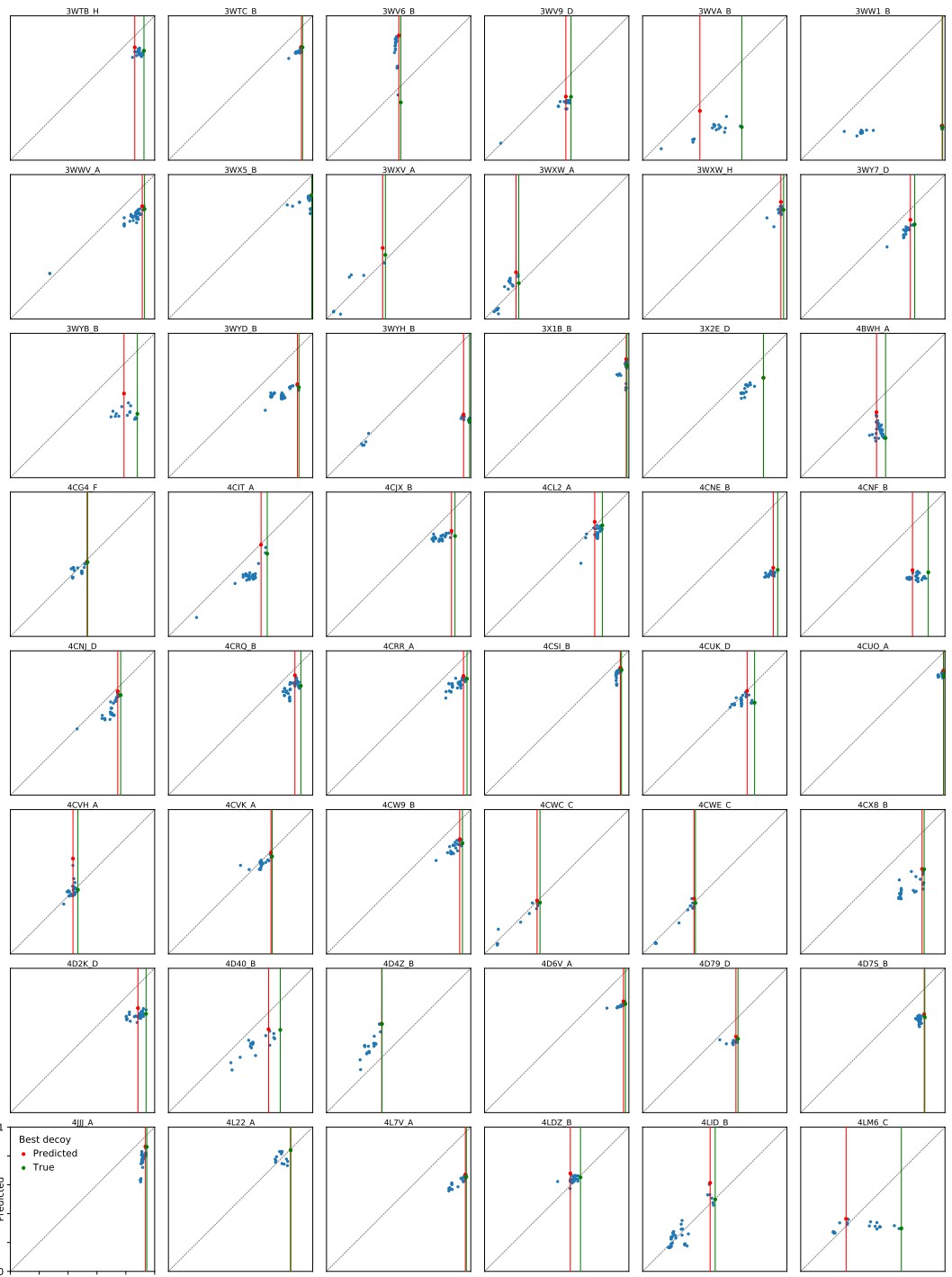

Figure 21: CAMEO: funnels (continues)

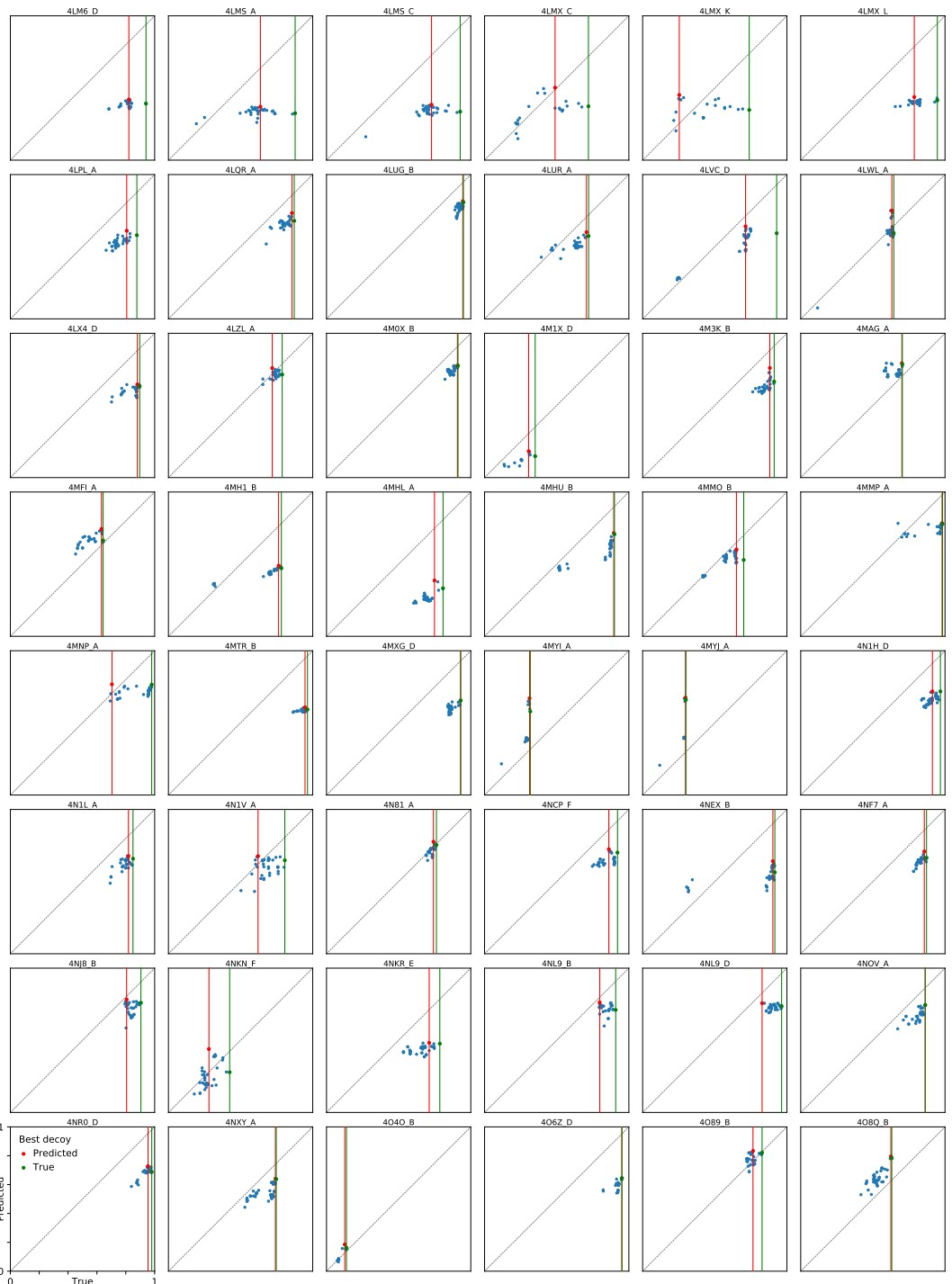

Figure 22: CAMEO: funnels (continues)

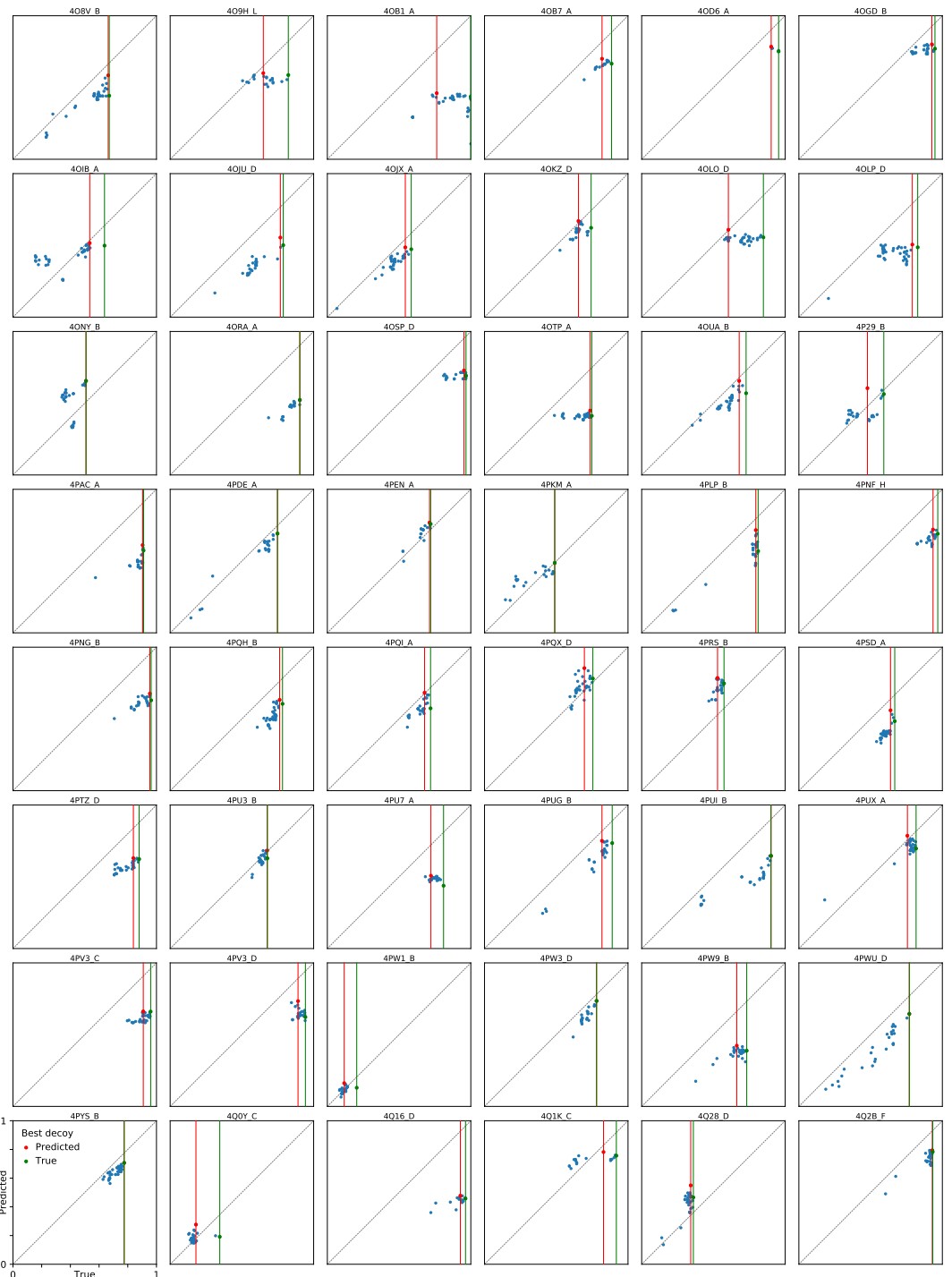

Figure 23: CAMEO: funnels (continues)

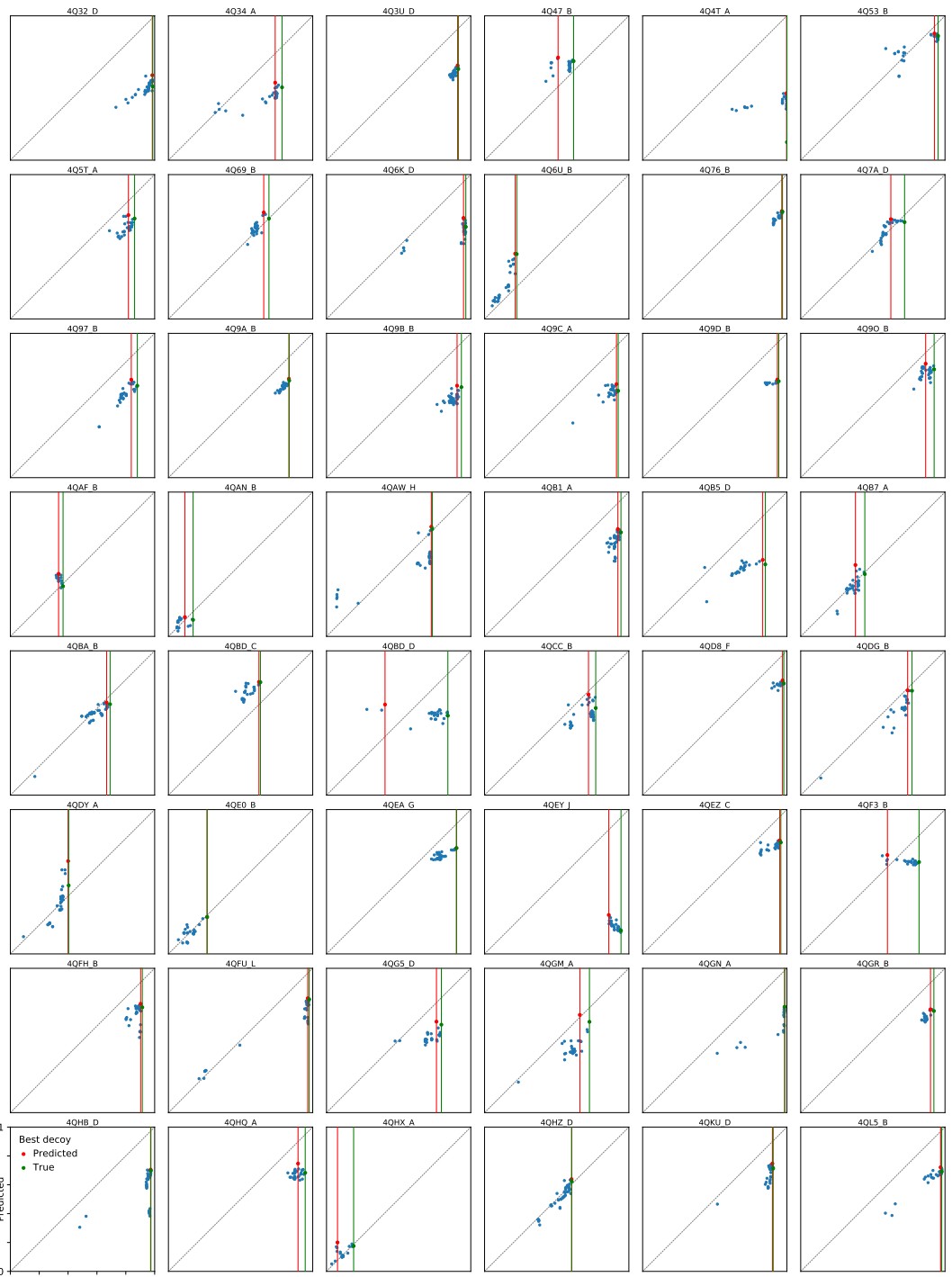

Figure 24: CAMEO: funnels (continues)

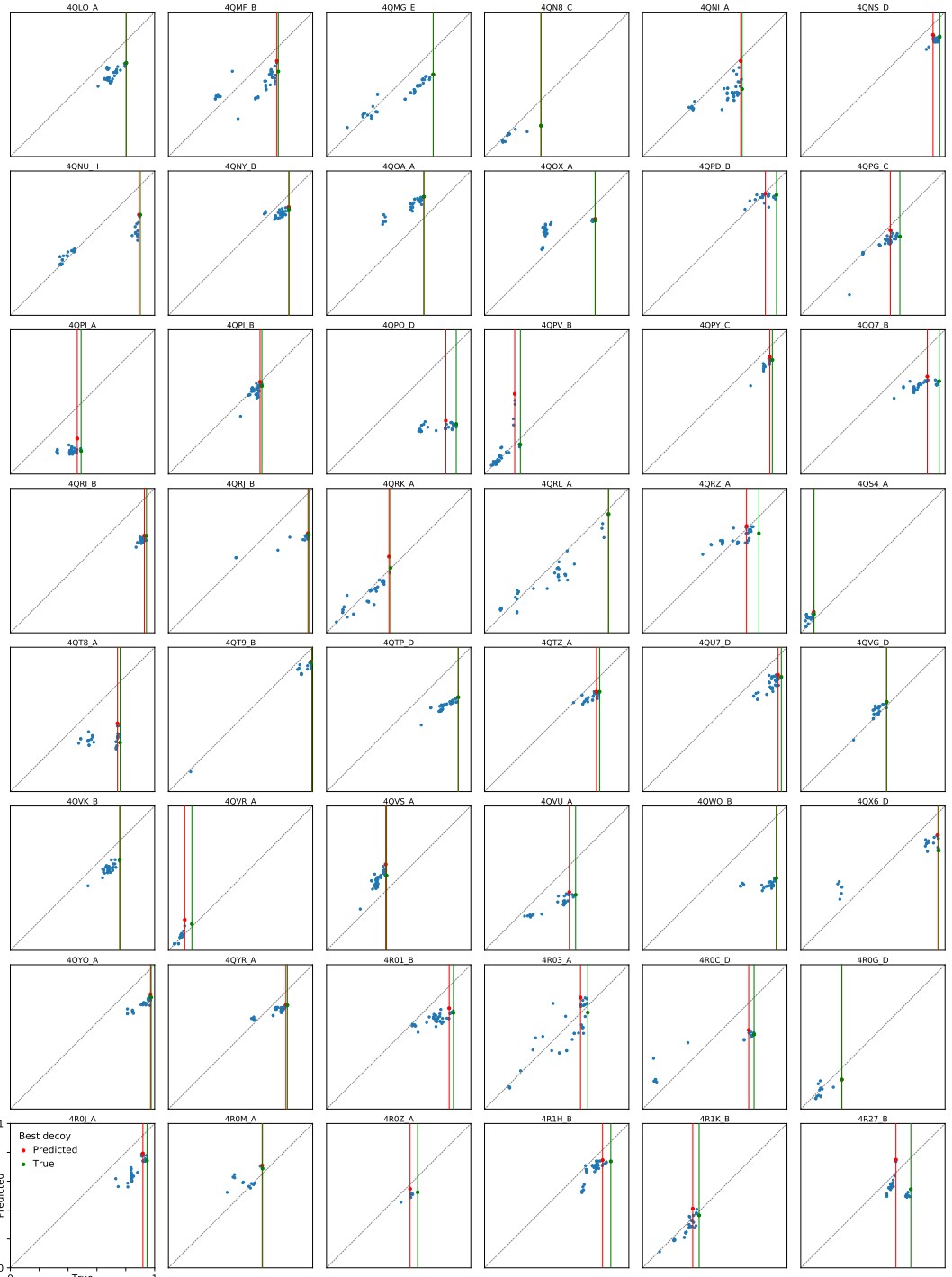

Figure 25: CAMEO: funnels (continues)

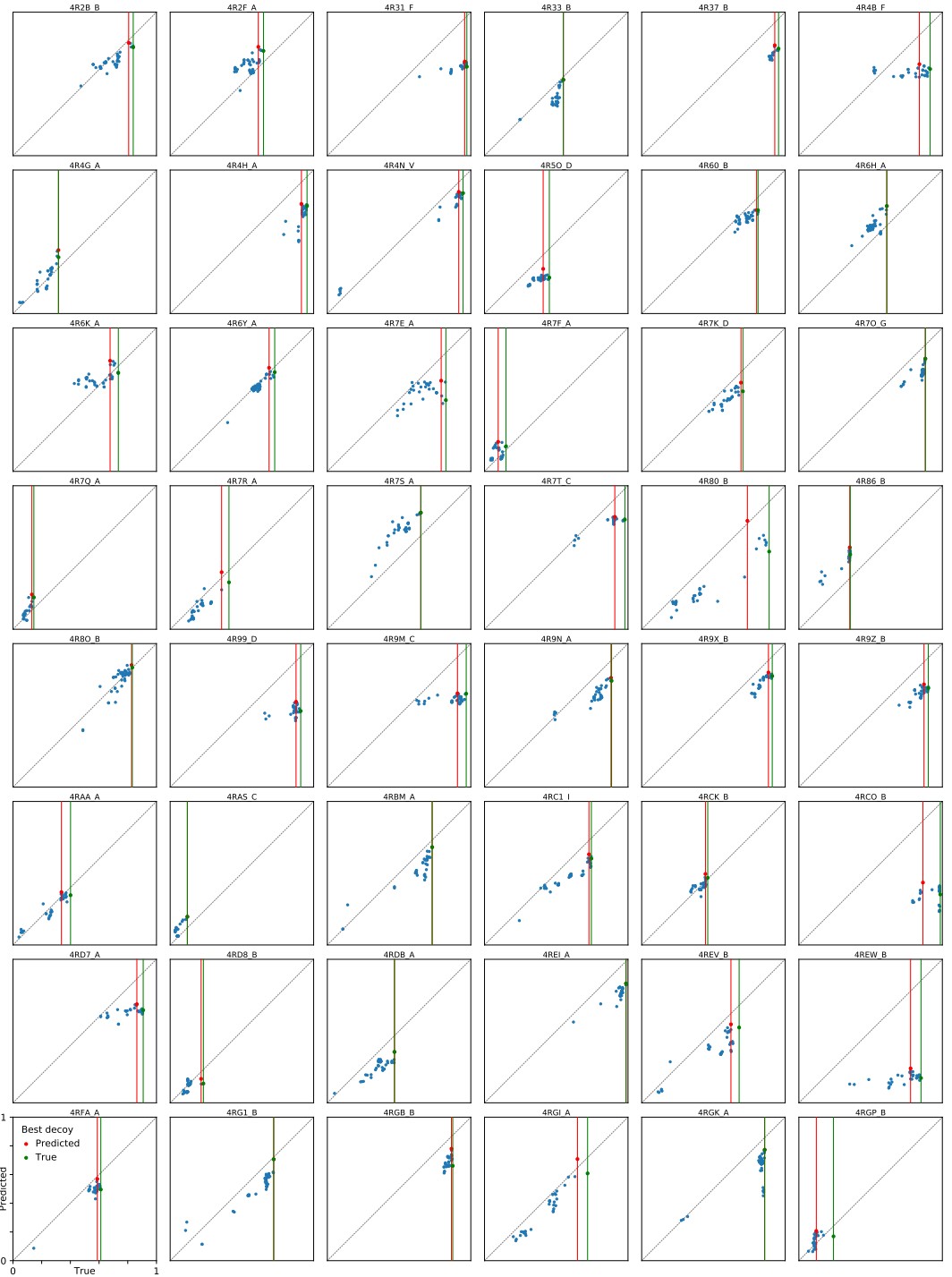

Figure 26: CAMEO: funnels (continues)

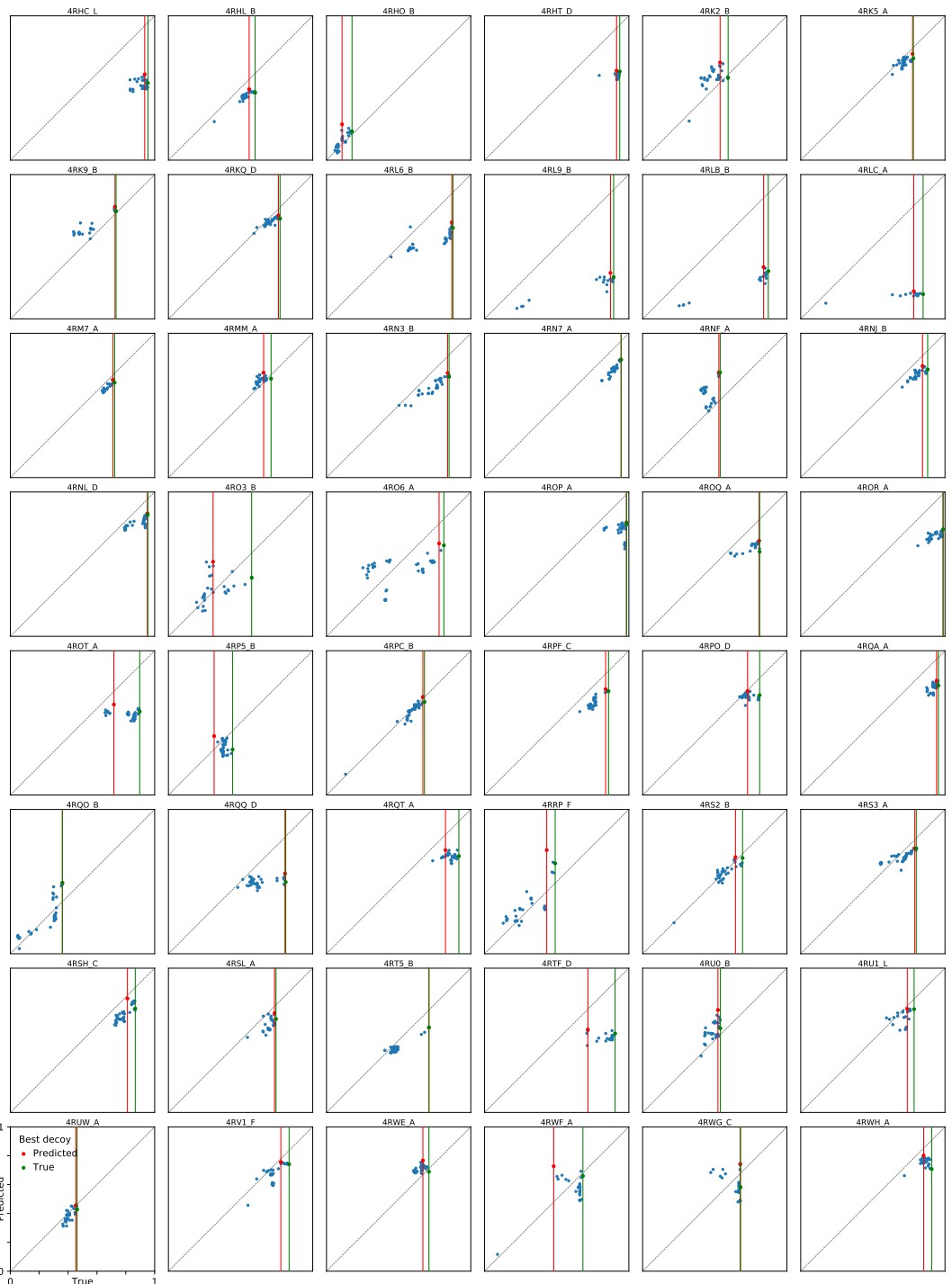

Figure 27: CAMEO: funnels (continues)

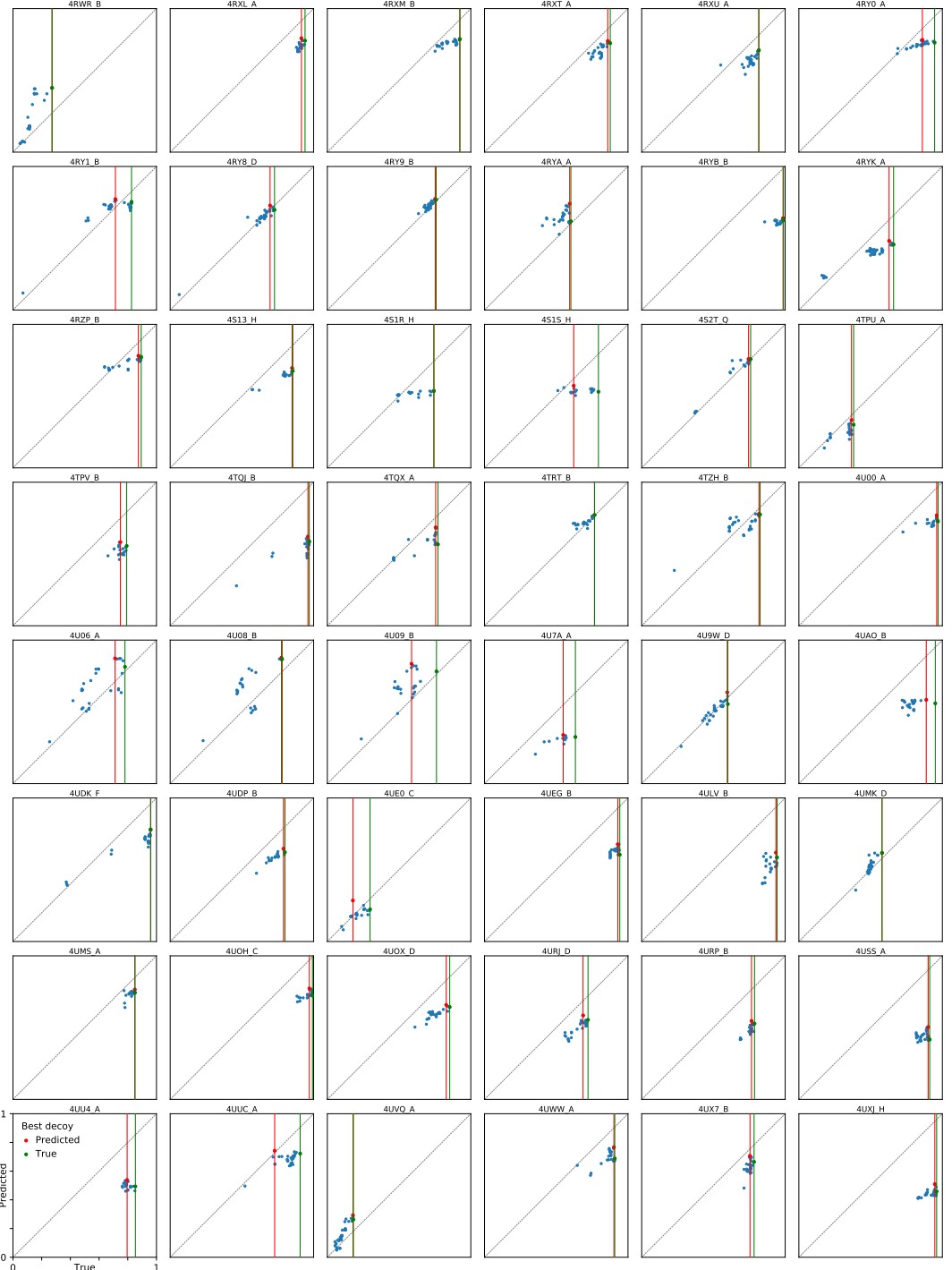

Figure 28: CAMEO: funnels (continues)

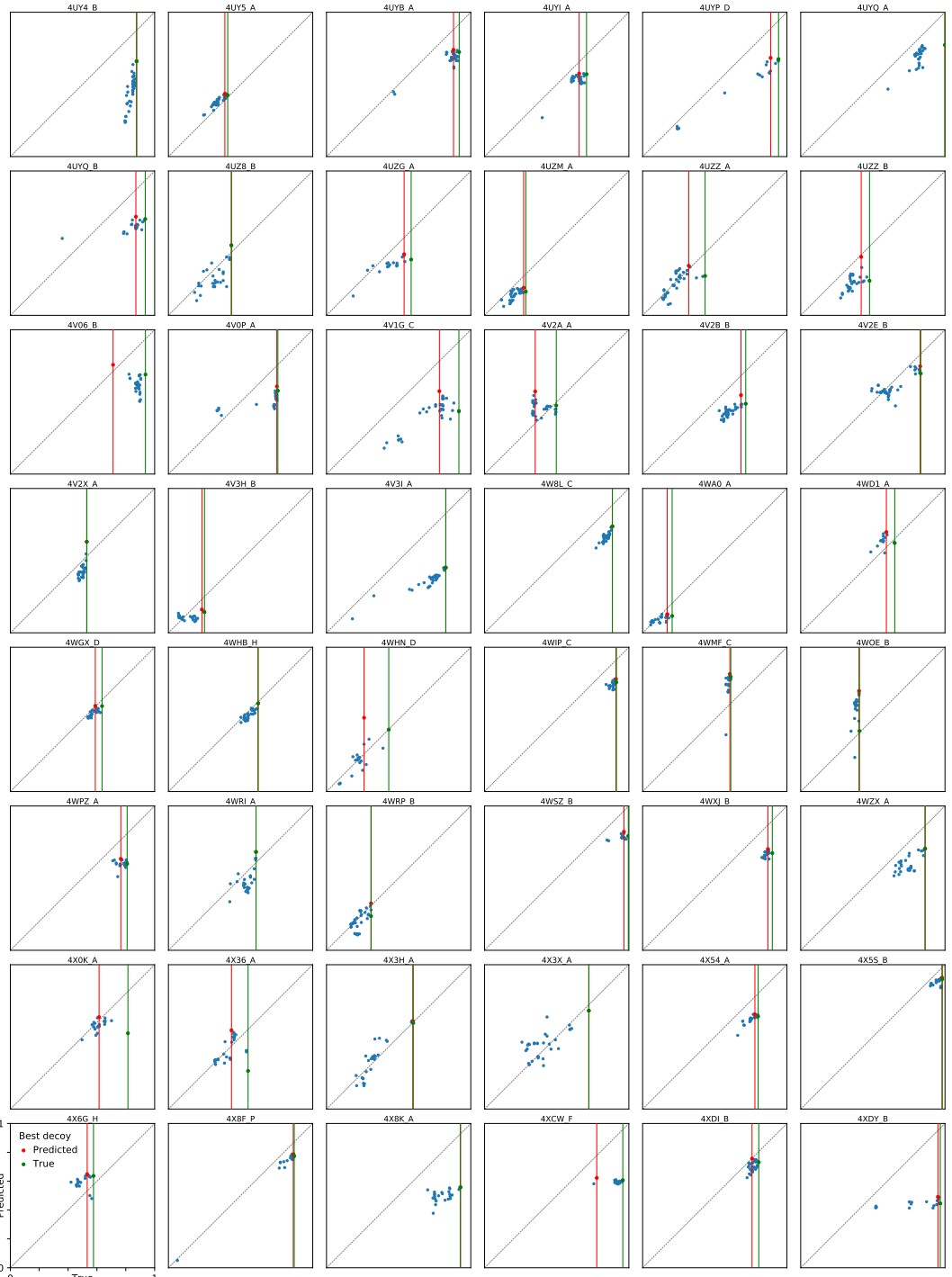

Figure 29: CAMEO: funnels (continues)

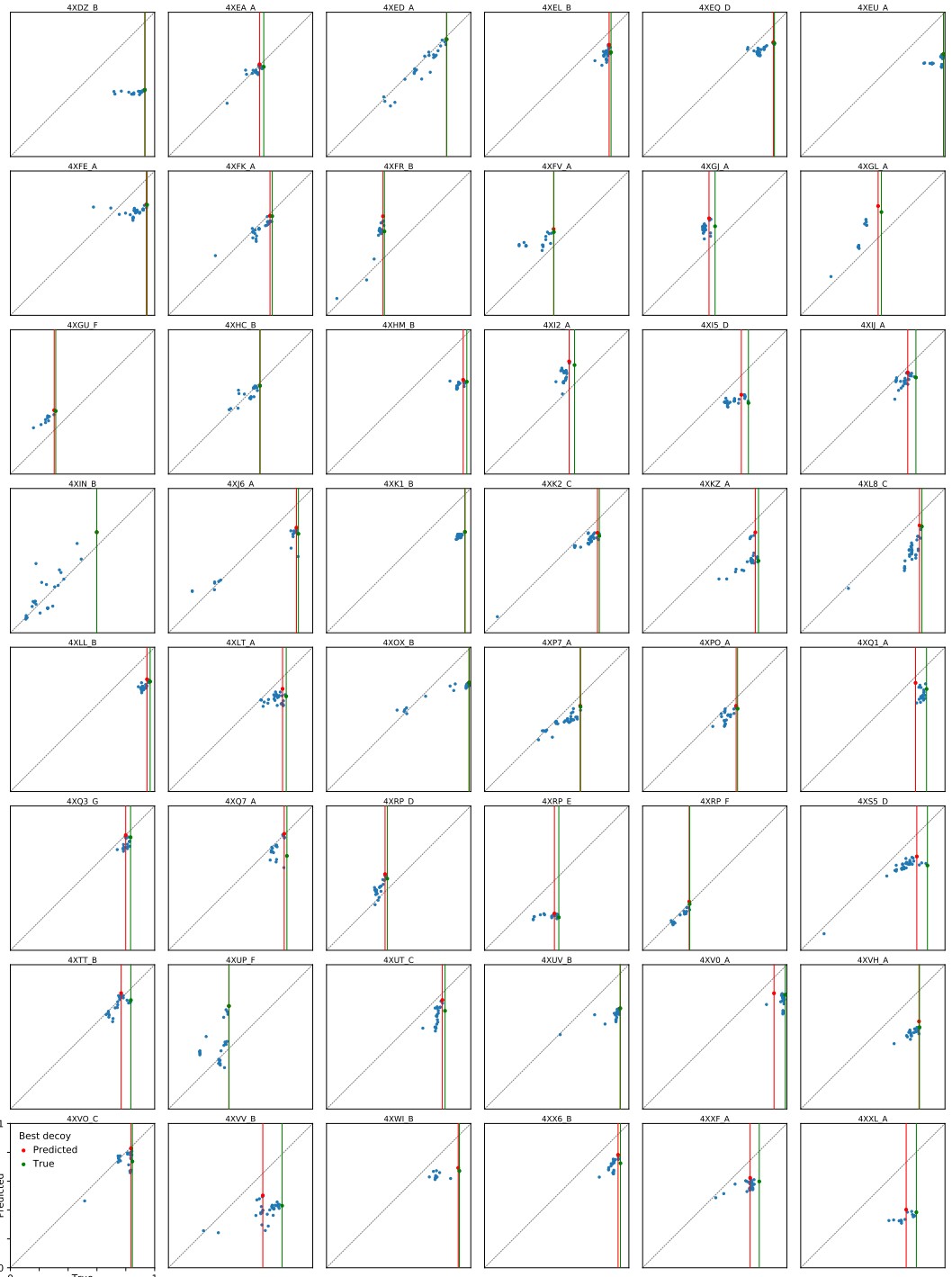

Figure 30: CAMEO: funnels (continues)

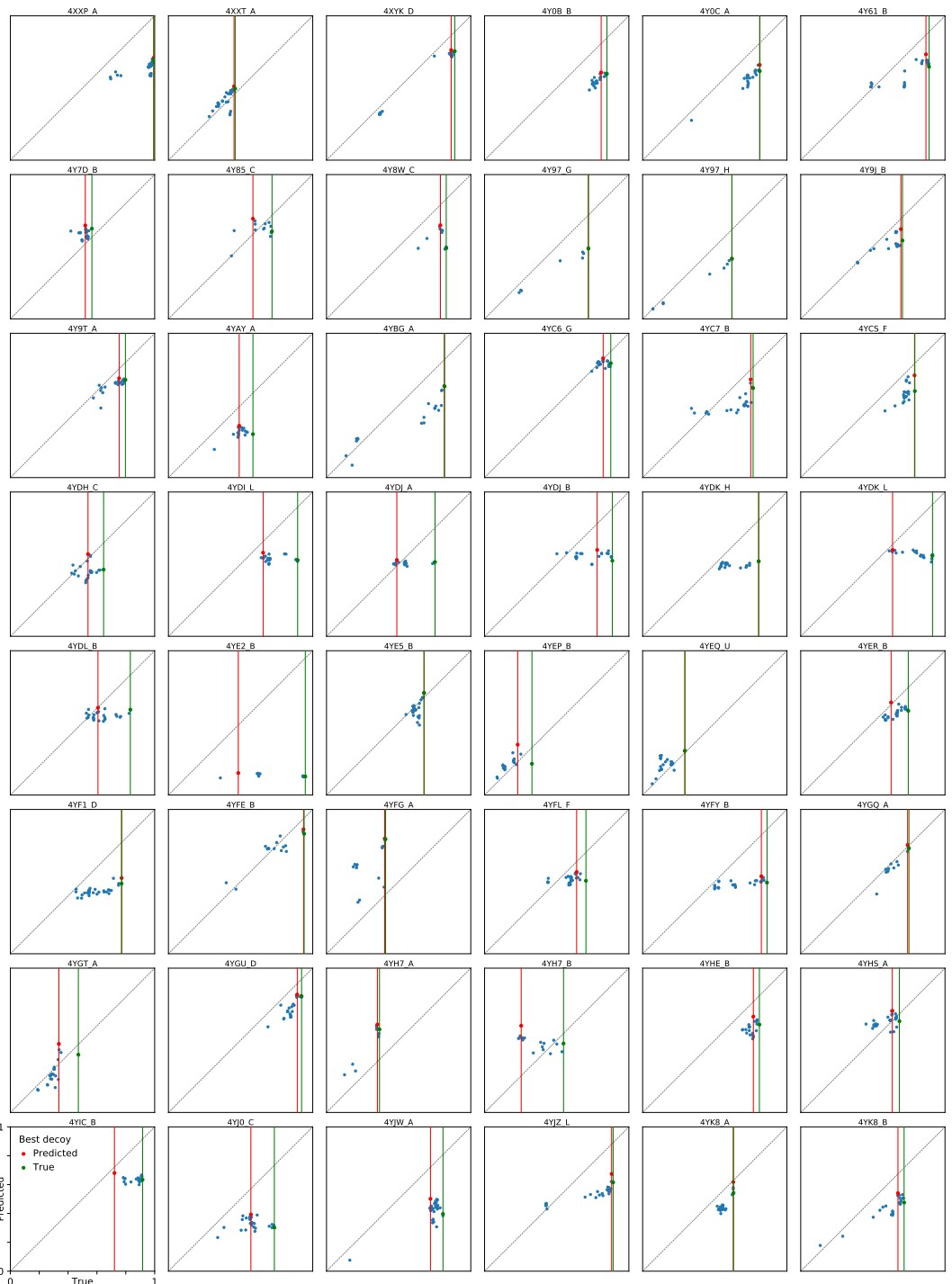

Figure 31: CAMEO: funnels (continues)

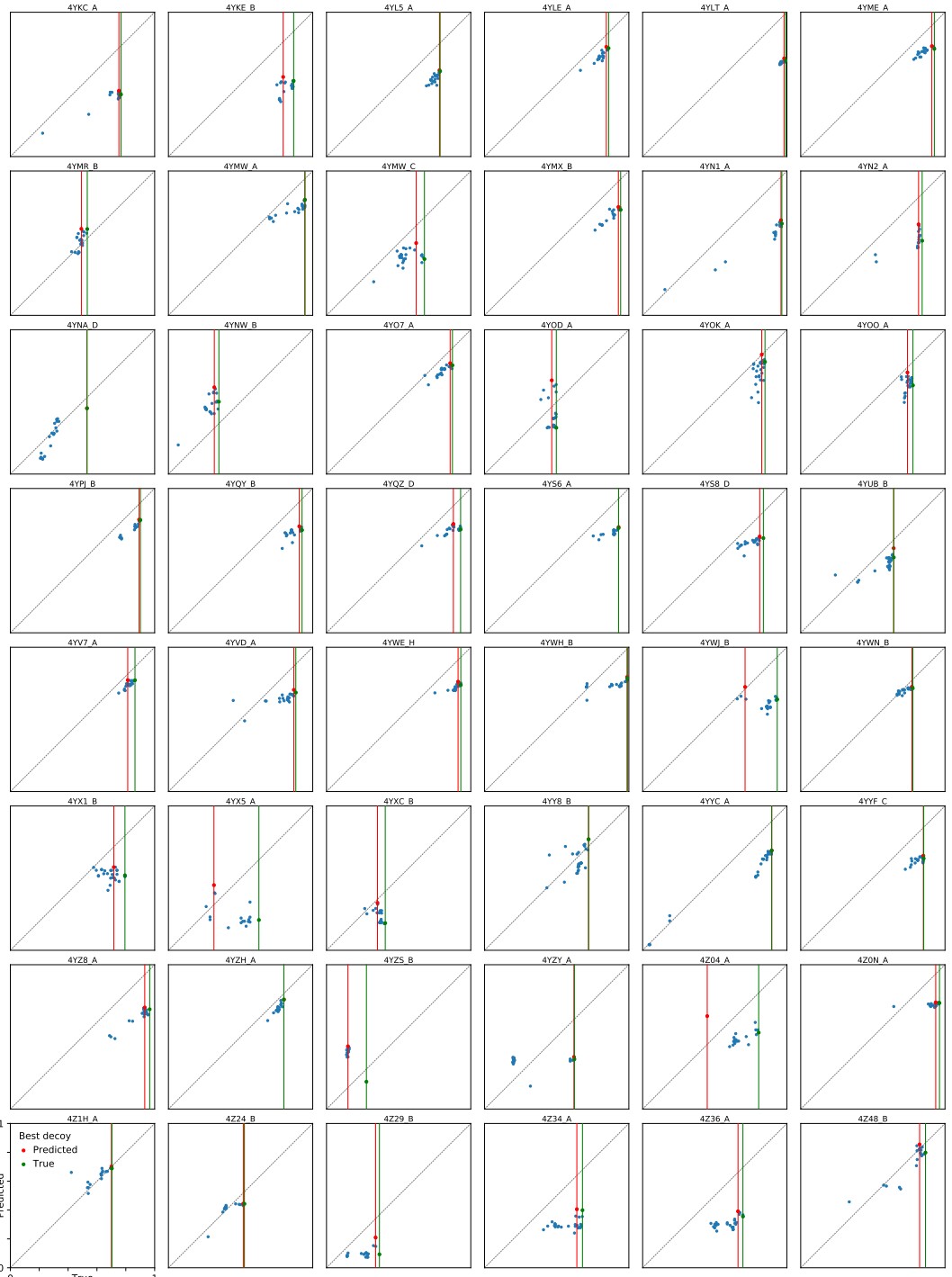

Figure 32: CAMEO: funnels (continues)

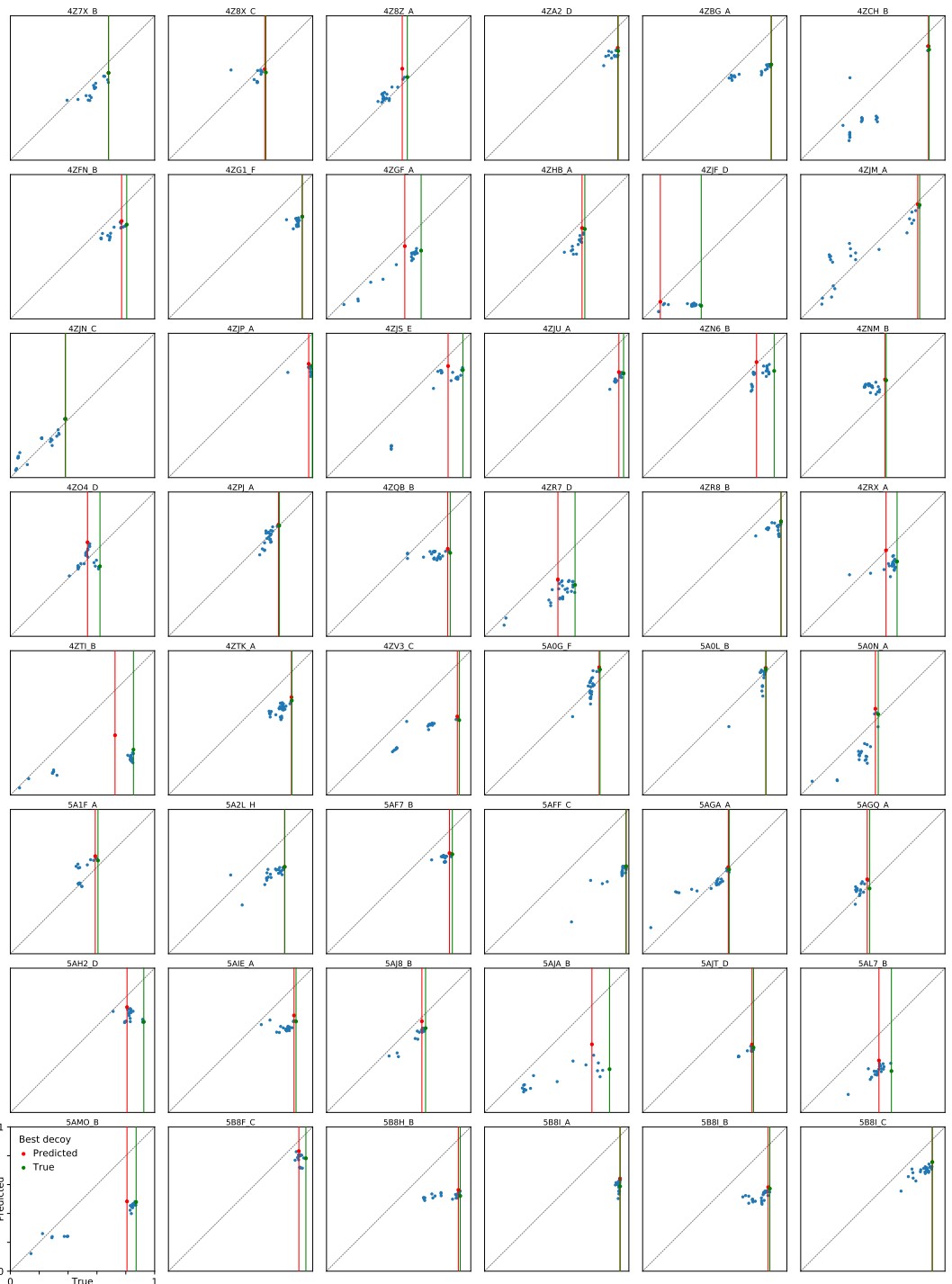

Figure 33: CAMEO: funnels (continues)

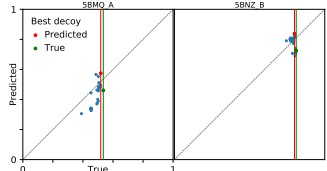

Figure 34: CAMEO: funnels (continued)

