# OpenReview forum: "GraphQA: Protein Model Quality Assessment using Graph Convolutional Network"
_ICLR.cc/2020/Conference — Reject_

### Official Review · AnonReviewer1 · 2019-10-24
**Official Blind Review #1**

**Rating:** 3

**Review:**

Proteins are sequences of Amino Acids. Identifying the 1-D sequence of a protein is straightforward. Each protein folds to a 3D structure. Determining the 3D structure of a protein (the protein folding problem) is expensive and hard.  It is well known that the function of a protein is determined by its 3D structure. Several computational methods have been proposed for protein structure prediction, but none of these models perform well in all circumstances. Different models perform well for different kinds of proteins. The current work deals with evaluating the different models to determine which model is likely to perform better on which protein.

The authors use Graph Convolutional networks with messaging to solve this problem of evaluating protein quality. Their method is evaluated using the Global Distance Test Total Score and Local Distance Difference Test (which is done at a residue level). They use several node and edge level features like DSSP, Partial Entropy and Self Intro.

Pros:

Their methods perform better than comparable methods using 1D and 3D CNNs. An ablation study is conducted to show the importance of various features. The paper is well written and the source code is provided for reproducibility. Overall, this is a good application paper showing the application of a known technique to solve a problem in a new domain.

Cons:

The novelty is minimal and the problem is of interest only to specialists in this domain.

**Experience Assessment:**

I have published one or two papers in this area.

**Review Assessment: Checking Correctness Of Derivations And Theory:**

N/A

**Review Assessment: Checking Correctness Of Experiments:**

I carefully checked the experiments.

**Review Assessment: Thoroughness In Paper Reading:**

I read the paper thoroughly.

---

> ### Author Response · Authors · 2019-11-13
> **Response to Reviewer 1**
>
> We thank the reviewer for carefully reading the paper and providing detailed comments that highlight its positive points. Here, we try to answer the two main concerns raised.
>
> _______________________________________________________________________
> 1) “The novelty is minimal”
>
> We understand the reviewer’s concern about technical novelty in general, but, in our opinion, this paper should be evaluated from an application perspective rather than for its technical novelty.
> We have minor *technical* novelties regarding 1) the graph representations we use and 2) the co-optimization of local and global scores. We certainly understand that these are not enough for a *technical* paper. However, as an *application* paper, our main novelty is to use a simple recent method (GCN) that is well motivated given the problem and that outperforms previous methods which have been developed for more than a decade. If our main strength had been the introduction of technical novelty, we would have followed a different evaluation strategy for a conclusive argument, mainly testing the method in different scenarios, while being less thorough in each experiment. On the other hand, as positively indicated by all reviewers, we chose to focus on a thorough analysis of this novel application, in terms of ablation studies of features and model components, multiple datasets, and different evaluation metrics.
>
> _______________________________________________________________________
> 2) “the problem is of interest only to specialists in this domain”
>
> We try to address this concern, shared with reviewer 2, from four different viewpoints.
>
> a) Most applications that are now commonly used as a benchmark in machine learning were initially niche domains, such as visual activity recognition, face verification, speaker recognition, sentiment analysis, etc. Our general argument is that “niche” is not synonymous with “unimportant” and that machine learning research should balance between pursuing state of the art on well-established benchmarks and expanding its horizons to less known fields.
>
> b) Although QA can be seen as a niche problem, it is actually an integrated part of most structure prediction pipelines and it has two major downstream applications. First, it can significantly increase the reliability of the predictions. As an example, one can look at the PconsFam database (pconsfam.bioinfo.se) where the structure is predicted for ~8000 Pfam families of unknown structure. Without QA it is virtually impossible to identify which of structures are correct, however, thanks to QA methods ~500 families can be assigned a >90% probability of being correct. Secondly, QA methods based on the evaluation of a single model are a key element in the development of end-to-end protein folding pipelines which are recently gaining popularity.
>
> c) In addition to the QA application presented here, we believe that our method and our representation can be transferred to other bioinformatics applications. First, as the graph representation relies only on the contact between residues (in contrast to the 1D and 3D representation), it should be straightforward to apply this method to the direct evaluation and refinement of contact maps. Given the recent improvement in learning-based contact (and distance) prediction, we foresee that an integration of our method with these algorithms could lead to important developments. Another task where our method could prove useful is protein-protein docking. Once again, the description of the problem as a graph is straightforward, but the task poses additional challenges since incorrect examples vastly outnumber the correct examples.
>
> d) The results can be interesting for an audience beyond BioInformatics. Graph Neural Networks (GNN) is a popular technique with several variants appearing in recent top machine learning conferences. This paper introduces a new graph-based benchmark for regression tasks. It introduces large datasets corresponding to the biannual CASP challenge and lays down a strong baseline for future development. Furthermore, the paper proposes new feature representations accompanied by thorough ablation studies, which can be useful for the general GCN audience, as pointed out by Reviewer 2.
>
> _______________________________________________________________________
>
> Finally, we thank the reviewer for evaluating this work as “a good application paper showing the application of a known technique to solve a problem in a new domain”. In that regard, we would also like to note that this is an application paper aligned with the ICLR call for papers that explicitly lists “computational biology” as a relevant application domain.

---

### Official Review · AnonReviewer2 · 2019-10-24
**Official Blind Review #2**

**Rating:** 6

**Review:**

The paper proposes use of graph convolutional networks (GCN) for quality assessments (QA) of protein structure predictions. In particular, protein structure prediction is a very active area of research witnessing steady progress during the previous decade or so. To estimate the quality of the prediction, many experimentally resolved structures are needed. However, experimental structure determination is expensive and many protein families are notoriously hard to experiment with. Thus, current estimates of the quality of the protein structure prediction models are incomplete and biased. As a result, there is an interest in guessing quality of protein structure predictions on protein families that are not characterized experimentally. Previous papers already proposed several neural network architectures for the QA task. This paper shows that GCN applied on a graph of a predicted protein structure achieves higher accuracy than the previously proposed neural networks

Strengths:
+ The proposed GCN outperforms other neural network baselines.
+ The protein representation is reasonable.
+ The paper is reasonably well written with a nice overview of the related work.
+ Ablation studies are well done, in clean graphs. This can be useful for other authors who work with GCNs.

Weaknesses:
- Methodological novelty is low -- this is a straightforward application of GCN
- The objective of QA is a bit suspect for the sole reason that the training and testing is performed using experimentally resolved protein structures. This data set is biased and there are no guarantees that the reported accuracy will hold over a vast range of protein families that are not structurally characterized.
- Separation encoding is done as a one-hot vector. This could probably be passed as a scalar value. Would be nice to have comparison between 1-hot vs scalar in the experimental results
- Formulas in section 2.3 are cryptic for audience unfamiliar with GCN and it is not specific to this application.
- Figure 3b) shows that there is a cluster of predicting 0 where the ground truth is bigger than 0.6.

Overall, this is a borderline paper. There is little methodological novelty and the QA application is a bit of a niche problem in bioinformatics. However, the results show a decent improvement over the state of the art in this particular application, so this paper might be of importance for a limited audience interested in this problem. Giving this work a benefit of the doubt the entered rating is a weak accept.

**Experience Assessment:**

I have read many papers in this area.

**Review Assessment: Checking Correctness Of Derivations And Theory:**

I assessed the sensibility of the derivations and theory.

**Review Assessment: Checking Correctness Of Experiments:**

I assessed the sensibility of the experiments.

**Review Assessment: Thoroughness In Paper Reading:**

I read the paper at least twice and used my best judgement in assessing the paper.

---

> ### Author Response · Authors · 2019-11-13
> **Response to Reviewer 2, part 1**
>
> We thank the reviewer for detailed and constructive feedback that definitely increases the quality of our work. Here we separately address the concerns raised.
>
> _______________________________________________________________________
> 1) “Methodological novelty is low -- this is a straightforward application of GCN”
>
> We understand the reviewer’s concern about technical novelty in general, but, in our opinion, this paper should be evaluated from an application perspective rather than for its technical novelty.
> We have minor *technical* novelties regarding 1) the graph representations we use and 2) the co-optimization of local and global scores. We certainly understand these are not enough for a *technical* paper.  However, as an *application* paper, our main novelty is to use a simple recent method (GCN) that is well motivated given the problem and that outperforms previous methods which have been developed for more than a decade. If our main strength had been the introduction of technical novelty, we would have followed a different evaluation strategy for a conclusive argument, mainly testing the method in different scenarios, while being less thorough in each experiment. On the other hand, as positively indicated by all reviewers, we chose to focus on a thorough analysis of this novel application, in terms of ablation studies of features and model components, multiple datasets, and different evaluation metrics.
>
> Finally, we thank the reviewer for noting that “results show a decent improvement over the state of the art in this particular application”. In that regard, we would also like to note that this is an application paper aligned with the ICLR call for papers that explicitly lists “computational biology” as a relevant application domain.
>
> _______________________________________________________________________
> 2) “The objective of QA is a bit suspect [...] using experimentally resolved protein structures”
>
> This is a thoughtful remark about the limitations of QA that transcends this work’s research question. Here we provide explanations as well as additional results to alleviate this concern.
> Our evaluation setup uses “old” datasets (CASP 7-10, 2007-2013) for training and the most recent datasets (CASP 11-13, 2015-2019 and CAMEO) for testing. As pointed out, all proteins in these datasets share a common factor that is instrumental for quantitative evaluations, namely that experimental determination of protein structure is feasible (e.g. they can be crystallized and scanned under an x-ray microscope). Other than this source of bias, which is explicit and outside our control, we believe that the scale and diversity of the targets considered for testing ensures a sufficient level of generalization. CASP 11, 12, 13 and CAMEO, in fact, portray a large spectrum of non-disordered proteins, e.g. ranging from very short to very long chains, from stand-alone to part of a complex, from hydrophobic to hydrophilic. A QA method that achieves good performances across these diverse datasets has the potential to correctly score computationally-modeled decoys of proteins whose true structure is unknown.
> Finally, as an additional study, we include a performance comparison between transmembrane and soluble proteins (section D.2). Predictably, GraphQA performs better on soluble proteins, which are more numerous in the training set, but it also scores transmembrane proteins to an acceptable degree.
>
> _______________________________________________________________________
> 3) “Separation encoding is done as a one-hot vector.”
>
> Thanks for bringing up this interesting question, it was definitely something worth looking into. There are actually two types of distances in play in the edges of our protein graphs: the spatial distance and the sequential distance (or separation). Our initial approach was to encode the spatial distance using a single RBF kernel and the separation as a categorical variable over biologically-motivated bins.
> As suggested by the reviewer, we conducted additional studies with several variants of this choice. For the spatial distance we tried: 1) removing it, 2) using the scalar value in Angstrom, 3) encoding the distance using 32 RBF kernels with unit variance. For the separation we tried: 1) removing it, 2) using the scalar separation (integer), 3) using a categorical encoding.
> Our findings, which are now included in the updated revision (section D.1), are the following. Categorical separation performs better than a scalar value for LDDT scores, while the effect on GDT_TS is minimal. For spatial distances, RBF encoding performs marginally better than the other two, both on LDDT and GDT_TS scores.

---

> > ### Author Response · Authors · 2019-11-13
> > **Response to Reviewer 2, part 2**
> >
> >
> > _______________________________________________________________________
> > 4) “Formulas in section 2.3 are cryptic for audience unfamiliar with GCN and it is not specific to this application”
> >
> > We agree with the reviewer that our implementation is not specific to protein quality assessment (we kept it general on purpose). However, by reviewing recent graph network literature, we noticed that the research community is far from reaching an agreement on what the standard formulation of a message-passing GCN should be (the graph-based submissions in this current edition of ICLR speak for themselves). This is not uncommon and has happened in the past with e.g. convolutional layers that nowadays we assume to be standardized. For this reason, we decided to briefly discuss the algorithmic implementation that we use in our method.
> > Section 2.3 is meant as a reference for the reader that is already familiar with GCN variants and had to be kept brief due to space constraints. In the same paragraph, we cite Battaglia et al., which serves as a reference for our implementation, and which we encourage to consult for a thorough investigation.
> >
> > To make the paper more accessible, we slightly modified section 2.3 and added a lengthy explanation of message-passing layers in the appendix section C.1, where all algorithmic steps are motivated and illustrated.
> >
> > _______________________________________________________________________
> > 5) “Figure 3b) shows that there is a cluster of predicting 0”
> >
> > That’s a keen observation. We double checked the predictions for individual targets and identified target T060 to be the problem. Figure 16 in the appendix clearly shows that the model defaults to predicting a small constant value for all decoys of T060, while predicting reasonable scores for all other targets. As a sanity check, we compared the plots of GraphQA with those of GraphQA-RAW, i.e. the model trained without self information, dssp and partial entropy features (in the repository they are located at results/allfeatures/CASP12/global_gdtts_funnel.pdf and results/residueonly/CASP12/global_gdtts_funnel.pdf respectively). It turns out that the model trained on “raw” amino acid features does not output the same degenerate predictions as its counterpart (the predictions are not perfect, but definitely better than a constant). We suspect that some error in the data pipeline might have produced misleading features for T060, e.g. the multiple sequence alignment program that extracts self information and partial entropy, or the DSSP program that computes secondary structure features. We added this remark to the paper.

---

> > > ### Author Response · Authors · 2019-11-13
> > > **Response to Reviewer 2, part 3**
> > >
> > >
> > > _______________________________________________________________________
> > > 6) “QA application is a bit of a niche problem in bioinformatics.”
> > >
> > > We try to address this concern, shared with reviewer 1, from four different viewpoints.
> > >
> > > a) Most applications that are now commonly used as a benchmark in machine learning were initially niche domains, such as visual activity recognition, face verification, speaker recognition, sentiment analysis, etc. Our general argument is that “niche” is not synonymous with “unimportant” and that machine learning research should balance between pursuing state of the art on well-established benchmarks and expanding its horizons to less known fields.
> > >
> > > b) Although QA can be seen as a niche problem, it is actually an integrated part of most structure prediction pipelines and it has two major downstream applications. First, it can significantly increase the reliability of the predictions. As an example, one can look at the PconsFam database (pconsfam.bioinfo.se) where the structure is predicted for ~8000 Pfam families of unknown structure. Without QA it is virtually impossible to identify which of structures are correct, however, thanks to QA methods ~500 families can be assigned a >90% probability of being correct. Secondly, QA methods based on the evaluation of a single model are a key element in the development of end-to-end protein folding pipelines which are recently gaining popularity.
> > >
> > > c) In addition to the QA application presented here, we believe that our method and our representation can be transferred to other bioinformatics applications. First, as the graph representation relies only on the contact between residues (in contrast to the 1D and 3D representation), it should be straightforward to apply this method to the direct evaluation and refinement of contact maps. Given the recent improvement in learning-based contact (and distance) prediction, we foresee that an integration of our method with these algorithms could lead to important developments. Another task where our method could prove useful is protein-protein docking. Once again, the description of the problem as a graph is straightforward, but the task poses additional challenges since incorrect examples vastly outnumber the correct examples.
> > >
> > > d) the results can be interesting for an audience beyond BioInformatics. Graph Neural Networks (GNN) is a popular technique with several variants appearing in recent top machine learning conferences. This paper introduces a new benchmark which naturally suits graphs. It introduces large datasets corresponding to the biannual CASP challenge and lays down a strong baseline for future development. Furthermore, as pointed out in the review, the paper proposes new feature representations accompanied by thorough ablation studies, which can be useful for the general GCN audience.

---

### Official Review · AnonReviewer3 · 2019-10-25
**Official Blind Review #3**

**Rating:** 3

**Review:**

This manuscript describes a new deep learning method for the prediction of the quality of a protein 3D model in the absence of the experimental 3D structure of the protein under study. The major idea is to model a protein 3D model using a graph. That is, each residue in a protein is modeled as a node and one edge is added to connect two residues if they are spatially close to each other.  Based upon this graph representation, the manuscript describes a graph convolutional neural network (GCN) to predict both local (i.e., per residue) and global quality. The authors showed that this GCN method works well on the CASP11 and CASP12 data. Unfortunately, there is no experimental result on CASP13 models, which significantly reduce my interest on this paper.

Minor concerns:

References are missing or misplaced at some places. For example, in the 1st sentence of the 4th graph, "While computational protein folding has recently received attention...", only protein design papers are cited. Some representative protein structure prediction papers shall be cited here.

**Experience Assessment:**

I have published in this field for several years.

**Review Assessment: Checking Correctness Of Derivations And Theory:**

I assessed the sensibility of the derivations and theory.

**Review Assessment: Checking Correctness Of Experiments:**

I assessed the sensibility of the experiments.

**Review Assessment: Thoroughness In Paper Reading:**

I read the paper at least twice and used my best judgement in assessing the paper.

---

> ### Author Response · Authors · 2019-11-13
> **Response to Reviewer 3**
>
> We thank the reviewer for carefully reading the paper and for suggesting action points to make bibliography more complete and our experiments more convincing for the audience. Here we address the two main concerns.
>
> _______________________________________________________________________
> 1) “no experimental result on CASP13”
>
> We did not include CASP 13 in the initial version since recently-published techniques use CASP 11/12 as a benchmark. Clearly, the recent CASP 13 represents an important dataset and is great of interest for many researchers. As suggested by the reviewer, we have now tested our model on publicly available targets of CASP13 and we report a comparison with other top participants of the challenge (section F.3). This comparison can be unfair since GraphQA is only trained on CASP 7-10, while other participants have likely (re)trained their models on all previous CASP datasets as well as other datasets. However, even without retraining, we achieve performances that are in line with the results presented for CASP 11 and 12.
>
> To strengthen our experimental evidence, we have also tested our model on the CAMEO dataset as well. Metrics and plots are reported in section F.4.
>
> _____________________________________________________________________
> 2) “References are missing or misplaced at some places.”
>
> We revised the mentioned paragraph to explicitly mention protein design and added additional references for structure prediction, namely:
> - “Distance-based protein folding powered by deep learning”
> - “High precision in protein contact prediction using fully convolutional neural networks and minimal sequence features.”
>
> We would be grateful if the reviewer could share any additional reference that is missing or misplaced so that we can further improve on this point.

---

### Author Response · Authors · 2019-10-07
**Errata corrige**

We would like to issue the following important corrections and apologize to the reviewers if the current version has caused any misunderstanding. They will be corrected in the next version.

Section 3.1, the last part of the second paragraph should read as:
Of these, we focus on R_target and R_model, which respectively measure the ability to rank decoys by quality and to distinguish the correctly-predicted parts of a model from those that need improvement. A description of these and other metrics can be found in appendix E.

Table 1, last row: GraphQA_RES refers to the GraphQA_RAW version mentioned in the text.

---

### Author Response · Authors · 2019-11-13
**To all reviewers**

We sincerely thank the reviewers for the thoughtful comments, which led us to further discuss our work and to improve the paper with additional experiments and explanations.

Here’s the list of updates in the revised version of the paper, for most of which we could only find room in the appendix due to the space constraint:
- Updated some references in the main text
- Additional test datasets, CASP13 and CAMEO (appendix F)
- Additional description of our GCN implementation (appendix C.1)
- Comparison of different representation for sequential and spatial distance (appendix D.1)
- Performance comparison of transmembrane vs soluble proteins in (appendix D.2)

For the sake of the followup discussions, we individually respond to the points raised by each reviewer. This means that our response can be sometimes repetitive (for overlapping concerns) across different reviewers. We apologize for this redundancy.

On top of the individual responses, we would like to comment on the relevance of our work. We suggest that our application paper should mainly be evaluated based on the following merits that it possesses:
a) the novelty of the method within the domain of the application,
b) the relevance of the paper to the venue,
c) the quality of the experiments,
d) the significance of the results.

We believe we cover these aspects as follows:
a) this is the first time that Graph Networks are used for protein model quality assessment,
b) “computational biology” is listed as a relevant application field in ICLR call for paper,
c) as indicated by reviewer 1 and 2, we provide thorough ablation studies of both feature representations and architectural components using several controlled runs per setup, which makes our experiments informative and reliable for follow-up works,
d) finally, as indicated by all three reviewers, our simple model works noticeably better than prior works, on several datasets, and according to various evaluation metrics.

To support this point, there are many application papers that have been considered relevant at top ML conferences for the aspects listed above. Here we report one representative paper each from the latest ICLR, NeurIPS and ICML conferences, that are closest to our work (without claiming that the opposite cases do not exist).

a) Similar OpenReview discussion regarding the technical novelty of a protein application paper, published at ICLR 2019: “Human-level Protein Localization with Convolutional Neural Networks”, https://openreview.net/forum?id=ryl5khRcKm
b) Graph Networks applied to a niche field, but with significant improvements and thorough analysis, published at ICML 2019: “Circuit-GNN: Graph Neural Networks for Distributed Circuit Design”, https://icml.cc/Conferences/2019/Schedule?showEvent=4826
c) Another niche application using a general gated graph recurrent network with noticeable performance published at NeurIPS 2019: “Devign: Effective Vulnerability Identification by Learning Comprehensive Program Semantics via Graph Neural Networks“, https://neurips.cc/Conferences/2019/Schedule?showEvent=14038

Furthermore, we argue that thoroughness and conclusiveness of experiments should outweigh technical novelties for an application paper. To support this, we kindly refer the reviewers to an ICLR2020 submission which raises concern around accepted papers that introduce technical novelties to GCNs but lack thorough experiments: “A Fair Comparison of Graph Neural Networks for Graph Classification”, https://openreview.net/forum?id=HygDF6NFPB

---

### Decision · Program_Chairs · 2019-12-19

**Decision:**

Reject

**Comment:**

This paper introduces an approach for estimating the quality of protein models. The proposed method consists in using graph convolutional networks (GCNs) to learn a representation of protein models and predict both a local and a global quality score. Experiments show that the proposed approach performs better than methods based on 1D and 3D CNNs.

Overall, this is a borderline paper. The improvement over state of the art for this specific application is noticeable. However, a major drawback is the lack of methodological novelty, the proposed solution being a direct application of GCNs. It does not bring new insights in representation learning. The contribution would therefore be of interest to a limited audience, in light of which I recommend to reject this paper.